# Integrative analysis of non-small cell lung cancer patient-derived xenografts identifies distinct proteotypes associated with patient outcomes

Shideh Mirhadi[1,2], Shirley Tam[3], Quan Li[3], Nadeem Moghal[3], Nhu-An Pham[3], Jiefei Tong[1], Brian J. Golbourn[4], Jonathan R. Krieger[5], Paul Taylor[1], Ming Li[3], Jessica Weiss[6], Sebastiao N. Martins-Filho [3,7], Vibha Raghavan[3], Yasin Mamatjan [3], Aafaque A. Khan [1], Michael Cabanero[3,7], Shingo Sakashita[3], Kugeng Huo[3], Sameer Agnihotri [4], Kota Ishizawa[8], Thomas K. Waddell [8,9], Gelareh Zadeh [3,9], Kazuhiro Yasufuku[8,9], Geoffrey Liu[3,10,11], Frances A. Shepherd[3,10], Michael F. Moran [1,2,12✉] & Ming-Sound Tsao [3,7,11,12✉]

Non-small cell lung cancer (NSCLC) is the leading cause of cancer deaths worldwide. Only a fraction of NSCLC harbor actionable driver mutations and there is an urgent need for patient-derived model systems that will enable the development of new targeted therapies. NSCLC and other cancers display profound proteome remodeling compared to normal tissue that is not predicted by DNA or RNA analyses. Here, we generate 137 NSCLC patient-derived xenografts (PDXs) that recapitulate the histology and molecular features of primary NSCLC. Proteome analysis of the PDX models reveals 3 adenocarcinoma and 2 squamous cell carcinoma proteotypes that are associated with different patient outcomes, protein-phosphotyrosine profiles, signatures of activated pathways and candidate targets, and in adenocarcinoma, stromal immune features. These findings portend proteome-based NSCLC classification and treatment and support the PDX resource as a viable model for the development of new targeted therapies.

[1] Program in Cell Biology, Hospital for Sick Children, Toronto, ON, Canada. [2] Department of Molecular Genetics, University of Toronto, Toronto, ON, Canada. [3] Princess Margaret Cancer Centre, University Health Network, Toronto, ON, Canada. [4] John G. Rangos Sr. Research Center, Children's Hospital of Pittsburgh, and Department of Neurological Surgery, University of Pittsburgh School of Medicine, Pittsburgh, PA, USA. [5] SPARC BioCentre, Hospital for Sick Children, Toronto, ON, Canada. [6] Department of Biostatistics, Princess Margaret Cancer Centre, Toronto, ON, Canada. [7] Department of Laboratory Medicine and Pathobiology, University of Toronto, Toronto, ON, Canada. [8] Division of Thoracic Surgery, Toronto General Hospital, University Health Network, Toronto, ON, Canada. [9] Department of Surgery, University of Toronto, Toronto, ON, Canada. [10] Department of Medicine, Division of Medical Oncology, University of Toronto, Toronto, ON, Canada. [11] Department of Medical Biophysics, University of Toronto, Toronto, ON, Canada. [12]These authors jointly supervised this work: Michael F. Moran, Ming-Sound Tsao. ✉email: m.moran@utoronto.ca; ming.tsao@uhn.ca

Lung cancer is the most frequently diagnosed and leading cause of cancer-related death worldwide, accounting for 13% of all cancers and 20% of all cancer deaths[1]. Non-small cell lung carcinoma (NSCLC) accounts for 85% of lung cancers, with adenocarcinoma (LUAD) and squamous cell carcinoma (LUSC) being the predominant histological types, accounting for approximately 60% and 30% of NSCLC cases, respectively[2,3].

Initial surveys of NSCLC genomes have led to the identification of oncogenic gene alterations, predominantly in LUAD, which have been particularly useful in guiding therapeutic decisions in advanced-stage NSCLC. Several targeted therapies are now approved for patients with specific genetic alterations in driver genes such as *epidermal growth factor receptor* (*EGFR*) kinase domain mutations and *anaplastic lymphoma kinase* (*ALK*) and *ROS1* rearrangements[4,5]. However, most NSCLC tumors lack actionable mutations. Identifying actionable targets for the majority of NSCLC, and especially those with more aggressive disease is a major motivation behind multi-omics profiling of NSCLC. Transcriptome-based molecular profiling of LUAD and LUSC have identified subgroups with prognostic value[6,7]. Multi-omic approaches have also been useful to identify the DNA alterations and methylation patterns associated with transcriptome subtypes[8–10]. However, despite these efforts, no actionable targets have been identified from these studies and our biological understanding of the heterogeneity between lung cancers remains quite limited.

In light of the limitations that have emerged from these genomic studies and the sometimes unpredictable relationships between DNA copy number, RNA expression, and protein levels, recent efforts have focused on mass spectrometry (MS)-based profiling of NSCLC proteomes[11–14]. While profiling of primary patient tumors provides invaluable information, it has limitations in not being able to readily distinguish between the biologic complexities of tumor epithelia and stroma, as both populations are molecularly indistinguishable in homogenized tissue samples. Thus, many novel biological insights may still be lacking for these cell populations. PDXs have emerged as powerful resources to overcome some of these limitations. In PDX models, human stroma becomes replaced by host murine components. Despite significant sequence conservation between the species, there is the potential to distinguish human tumor epithelial and murine stromal proteins based on recognition of species-specific polypeptide sequences. PDXs, including from NSCLC, also provide a resource enriched in the more aggressive forms of cancer,

where there is the greatest urgency to develop clinically meaningful insights. NSCLC patients whose resected tumors engraft to form a PDX have worse overall survival than patients whose tumors fail to engraft[15–18]. The engrafted patient tumors retain the phenotypic features of the primary tumors, including histology, mutational landscape, RNA and protein expression[19–24]. We previously measured high correlations ($r_s > 0.67$) between 11 PDX and their matched patient NSCLC for individual profiles of their DNA copy number, mRNA and protein abundances[25]. In another study comparing 36 matched PDX-primary NSCLC we demonstrated retention of >90% of SNP mutations and the close recapitulation in PDX models of gene expression, methylation, and protein-phosphotyrosine (pY) profiles[26]. These observations indicate a high degree of conservation of molecular features between primary and PDX tumors, suggesting they are valid models suitable for pre-clinical studies.

In this study, we establish a large cohort of 137 NSCLC PDX models and profile gene expression, gene copy number variation, DNA methylation, exome mutations, proteome and pY-proteome (Fig. 1). DNA copy number variation and exome sequences of PDXs represent that of primary tumors. Gene expression and methylation signatures defined in primary tumors effectively stratify the PDXs. Unsupervised MS-based proteome analysis identifies 3 LUAD and 2 LUSC proteotypes, which are associated with distinct pY-proteome profiles and patient survival. Interrogation of the proteotypes reveals features of active pathways. Proteotype markers effectively stratify unrelated cohorts[12,14] of primary tumors according to proteotype. Proteotype signatures also effectively stratify Cancer Dependency Map (DepMap) NSCLC cell lines in which gene knockout/down sensitivities aligned with proteotype-defined candidate targets[27,28] (Fig. 1). Two of the identified LUAD proteotypes have distinctive stromal proteomes, which suggests that distinct molecular signaling by these proteotypes specifies tailored stromal microenvironments. These insights into lung cancer biology suggest that the further characterization of NSCLC proteotypes in NSCLC models including our PDX collection may be an approach to test emerging therapeutic hypotheses.

## Results

**Generation of Stable NSCLC PDX Models.** A total of 500 NSCLC patient samples, 443 primary resected and 57 endobronchial

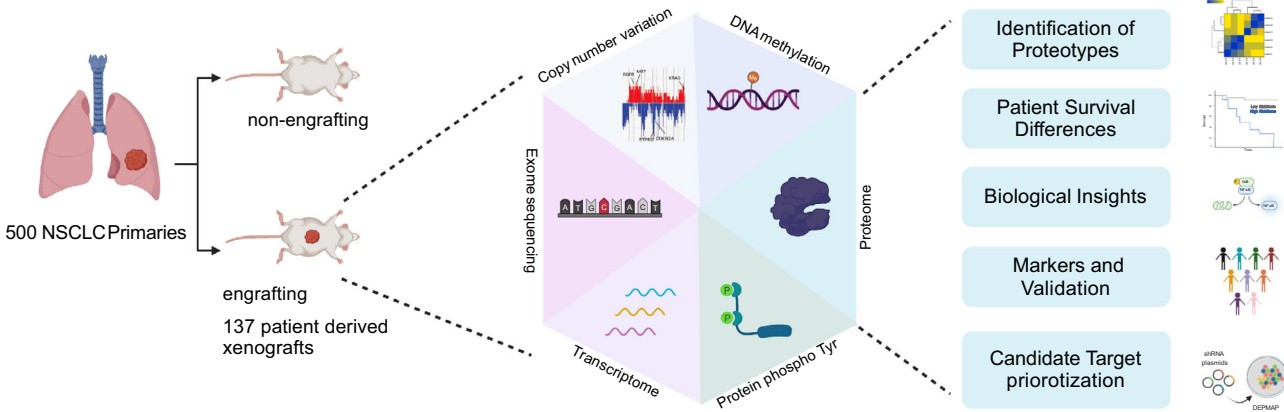

**Fig. 1 A roadmap to cancer proteotype discovery and utility.** A subset of 137 of 500 primary NSCLC tumors engrafted to yield PDX models. PDXs represent the most aggressive subset of NSCLC and were profiled for gene expression, gene copy number variation, DNA methylation, exome mutations, proteome and phosphotyrosine(pY)-proteome. Proteome profiling revealed proteotypes associated with patient survival differences. Proteotypes display distinctive active pathway features and associated candidate therapeutic targets. Signatures comprising proteotype markers effectively stratify orthogonal NSCLC primary tumors[12,14] as well as NSCLC DepMap cell lines[25], which enables a degree of candidate target validation and prioritization based on alignment with DepMap sensitivities[26].

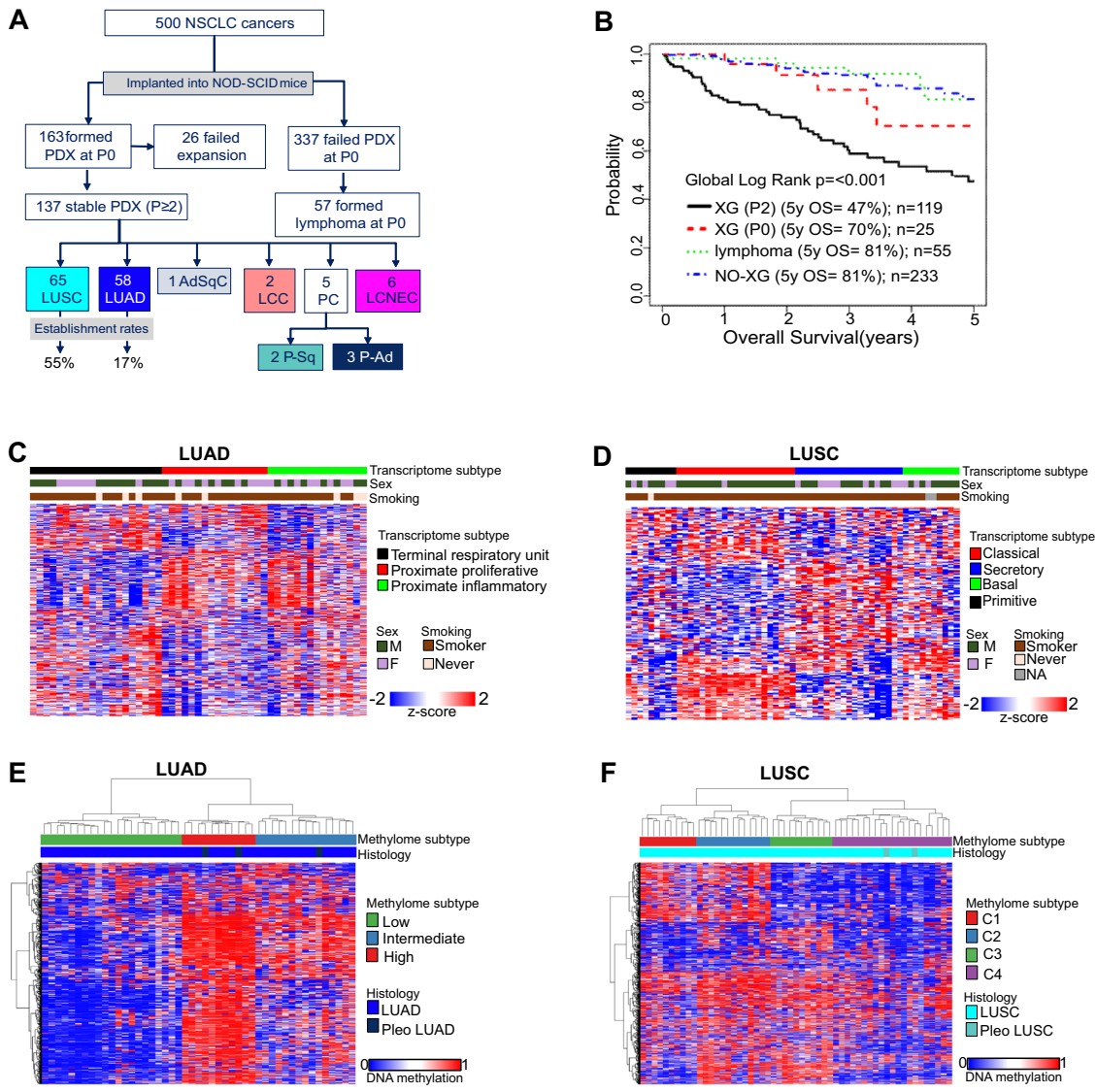

**Fig. 2 NSCLC PDX models represent primary tumor transcriptome and methylome features. A** Workflow for generation of NSCLC stable PDX models including primary resections and EBUS recurrences. See also Supplementary Table 1. **B** 5-year survival Kaplan–Meier plot of primary resected cases. **C** Assignment of PDXs to the 3 TCGA described LUAD transcriptome subtypes. See also Supplementary Fig. 1. **D** Assignment of PDXs to the 4 TCGA described LUSC transcriptome subtypes. See also Supplementary Fig. 1. **E** Identification of three major methylation groups among LUAD and **F** four major methylation groups among LUSC using unsupervised clustering of top 4000 most variable CpGs (in promoter region or within 1500 bases of the transcription start site).

ultrasound (EBUS) biopsy from recurrent cases, were implanted into non-obese diabetic severely compromised immunodeficient (NOD-SCID) mice (Fig. 2A; Supplementary Table 1). This resulted in formation of 163 PDX at P0 (passage 0 designates initial implant with patient tumor fragment). A subset of 137 PDX tumors were successfully serially passaged at least three times (P2), while 26 were unstable beyond P0. Thus, the overall engraftment rate for stable PDX formation was 27% (137/500). The engraftment rate was higher for resected specimens (125/443; 28%) than EBUS samples (12/57; 21%) (Supplementary Table 1). Consistent with previous reports, the engraftment rate was higher for LUSC (55%, 65/119) than LUAD (17%, 58/336)[15,29]. Of the 500 implanted patient tumors, 57 (11%) formed lymphoma in P0 mice. We reported previously that P0 lymphomas were of human origin and likely represent transformed tumor infiltrating lymphocytes[30]. From the resected cohort, the ability of patient tumors to form a stable PDX is associated with significantly worse overall patient survival (OS) (Fig. 1B). Non-stable engrafting tumors (XG, P0) had the second

worst OS followed by non-engrafting tumors (NO-XG). Lastly, patients with engraftment outgrowth as lymphoma had similar OS as that of patients with non-engrafting tumors (Fig. 2B). Clinical and demographics of patients contributing to PDX generation are summarized in Supplementary Tables 2 and 8.

**NSCLC PDX models represent primary tumor transcriptome and genome features.** LUAD and LUSC were characterized by gene expression subtypes[6,7] that were validated in datasets of The Cancer Genome Atlas (TCGA)[9,10]. The TCGA signature algorithms were applied to our PDX gene expression profiles, the 3 known subtypes of LUAD (*Terminal Respiratory Unit*, *Proximate Proliferative* and *Proximate Inflammatory*) and 4 LUSC subtypes (*Classical, Secretory, Basal* and *Primitive*) (Fig. 2C, D). The PDX models represent all of the transcriptome-based subtypes in comparable proportions to patients in TCGA cohorts[6,31] (Supplementary Fig. 1A, B). Principal component analysis (PCA) of

the methylation profiles of 47 LUAD and 55 LUSC PDX models showed strong separation of tumors into histological subtypes (Supplementary Fig. 1C). TCGA methylation datasets using the top 4,000 most variable CpGs and unsupervised clustering indicated 3 LUAD and 4 LUSC methylation groups[9,10]. Using the same strategy, we identified 3 LUAD and 4 LUSC subgroups among the PDXs (Fig. 2E, F). The data suggests that DNA methylation signatures are also conserved and represented in the PDX models. Although the frequency of methylation-based subtypes was on par with that of primary LUAD, in LUSC the C1 subtype was under-represented by 50% and the C2 subtype was over-represented by 100% in comparison to a patient population[13]. These suggest that biological aspects related to these subtypes might influence engraftment.

To examine the frequency of DNA alterations, PDX models were profiled for mutations and copy number. We observed that alterations frequently identified in LUAD and LUSC primary tumors are represented in the PDX models but in some cases were over- or under-represented compared to the frequencies seen in patient tumor populations[32] (Fig. 3A, B) (Supplementary Data 1). For instance, KRAS mutations are over-represented in

LUAD PDXs at 50% frequency compared to 30% in patient primary tumors, whereas EGFR sensitizing mutations in exon 18–21 are under-represented[32] (Fig. 3A; Supplementary Data 1). These findings are consistent with mutant KRAS being associated with better and mutant EGFR poorer engraftment rates, respectively, and poorer and better prognoses, respectively[9,31,33]. NFE2L2, FAT1 and NOTCH1 mutations were over-represented in LUSC PDXs compared to primary tumors[32], suggesting such alterations might favor a more aggressive cancer phenotype and PDX formation. CNV in LUAD and LUSC PDX tumors mirrors closely primary tumors, (Fig. 3C, D; Supplementary Data 1). Overall, these analyses revealed that the NSCLC PDX models retain genomic features that resemble primary tumors including some that may be related to aggressiveness and engraftment.

**Proteome profiling of NSCLC PDX tumors.** Tandem mass tag (TMT)-based quantitative MS analysis of PDX tumors was undertaken (Supplementary Fig. 2A). For data quality assurance, a replicate sample pair in the same experimental group and two pairs of replicates samples split into different experimental groups

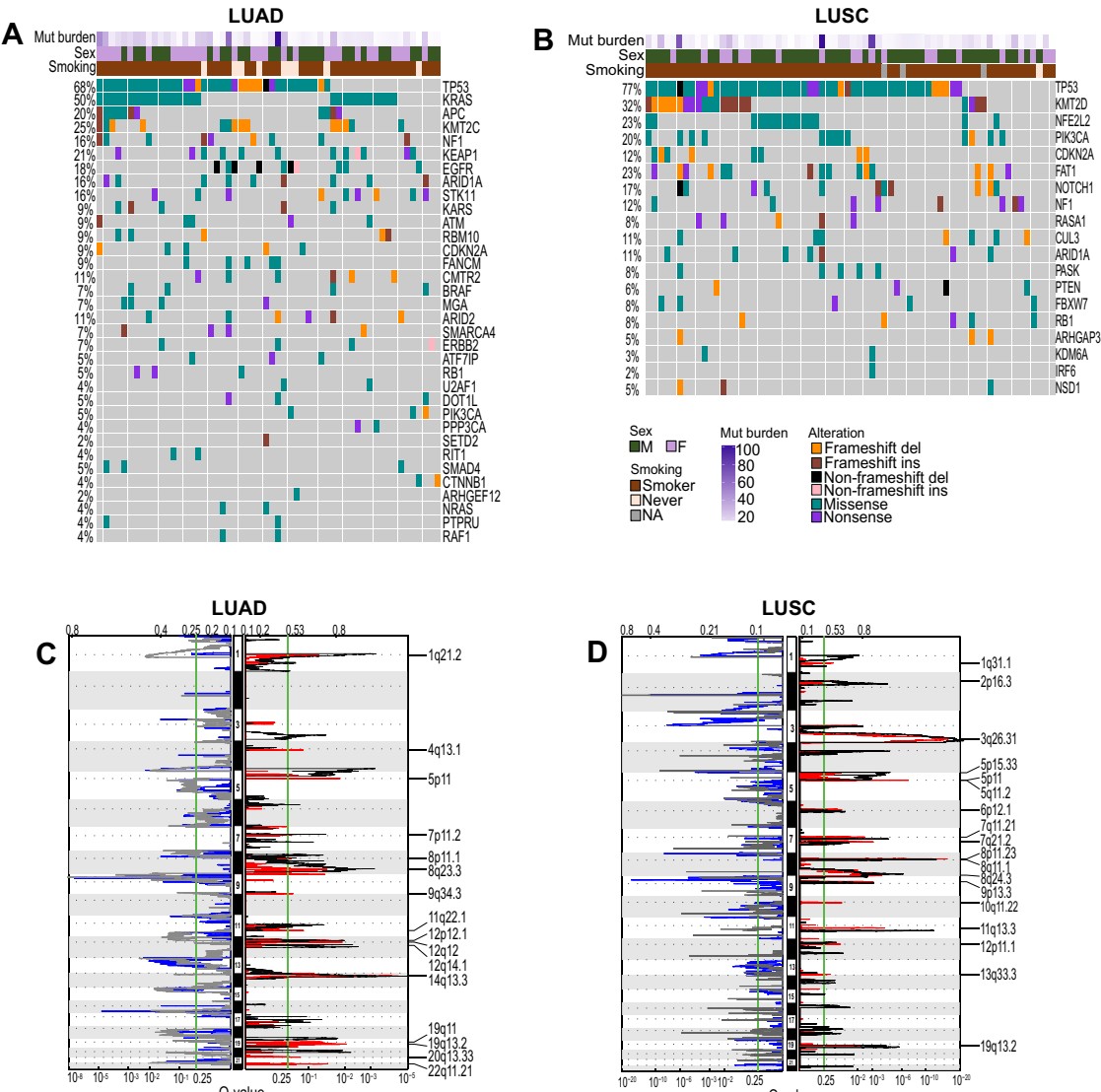

**Fig. 3 NSCLC PDX models represent primary tumor genomic features. A**, **B** Oncoprint showing DNA alterations in frequently altered genes of **A** LUAD and **B** LUSC. **C**, **D** LUAD (**C**) and LUSC (**D**) PDXs GISTIC showing significantly amplified regions in red and significantly deleted regions in blue, TCGA CNV is overlaid for comparison with black showing significantly amplified and gray showing significantly deleted CNV regions in LUAD TCGA patients.

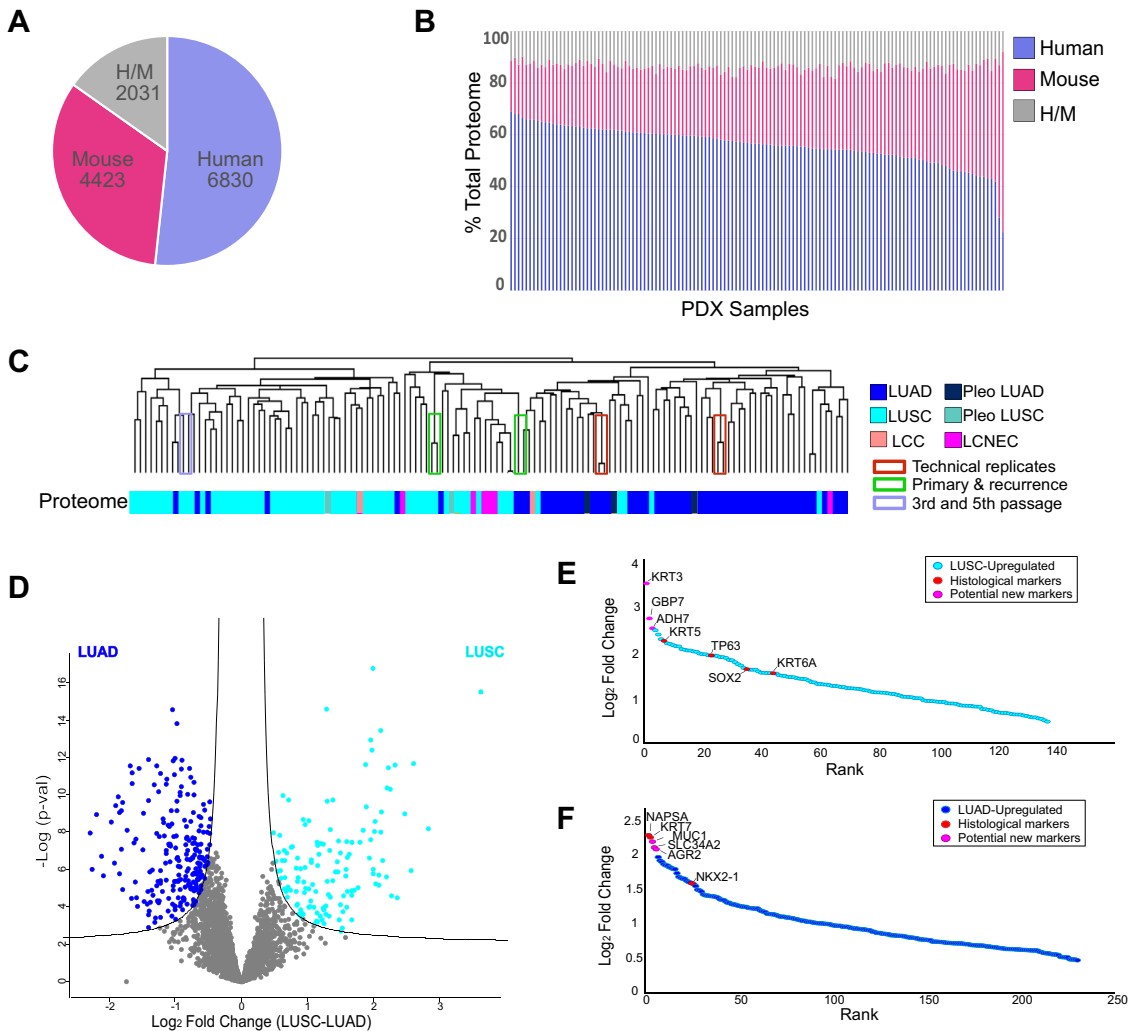

**Fig. 4 Proteome measurement of PDX models. A** Pie chart showing 6830 human (tumor), 4423 mouse (stroma) and 2031 ambiguous human/mouse proteins detected by proteome analysis. **B** Tumor/stromal composition of each PDX sample shows wide range of tumor vs. stroma content. **C** Histological separation by unsupervised hierarchical clustering of PDXs by tumor proteome. **D** Differential proteome between histological types LUAD and LUSC (two-sided *t*-test, FDR < 0.001). **E**, **F** Significantly differential upregulated proteins (FDR < 0.001) are ranked based on fold-change (**E**) for LUSC and (**F**) LUAD.

were analyzed. The technical replicates provided a readout for fidelity of the normalization method and technical robustness. A strong linear relationship between the replicates was seen for each pair ($R^2 \geq 0.94$) (Supplementary Fig. 2E). PCA verified that samples did not cluster based on experimental group or isobaric labels (Supplementary Fig. 2F, G). MS analysis of 133 PDX samples uncovered a total of 13284 proteins using a strict false discovery rate (FDR) of 0.01 of which 6830 were identified as human, 4423 as mouse, and 2031 that did not contain unique human or mouse peptides and therefore were assigned as human/mouse (Fig. 4A). To assess tumor-stroma composition, the fraction of total ion intensity corresponding to human, mouse, and human/mouse proteins was determined for each PDX sample (Fig. 4B) (Supplementary Data 1). This provided an opportunity to correct for discrepancies in tumor (i.e., human) cell composition across samples, which ranged between 20–70% (Fig. 4B; Supplementary Data 1). This ensured that measured changes in protein abundance reflects proteome remodeling in tumor cells and not differences in tumor cellularity.

To address the postulate that the proteome is largely unpredictable based on abundance of genes and transcripts, pairwise correlations were made between CNV, mRNA and protein[25]. These analyses were applied for genes/gene products represented at the

protein level, and by using only human proteins that were quantified in all samples. The resulting Spearman's Rho values were positive but low in magnitude. The median Spearman's Rho was 0.33 for CNV-mRNA, 0.22 for CNV-Protein, and 0.3 for mRNA-Protein (Supplementary Data 1; Supplementary Fig. 2B–D).

Unsupervised hierarchical clustering based on human/tumor proteins identified in at least 70% of 133 PDX tumors revealed three major clusters: one comprised mainly LUSC, a second comprising LUSC together with most of the large cell neuroendocrine carcinoma (LCNEC) samples, and a third comprised of mainly LUAD (Fig. 4C). Technical replicates, primary/recurrent pairs, and different PDX passages from the same primary sample consistently clustered together (Fig. 4C). These results suggest a high degree of consistency and accuracy in the proteomic platform, and minimal proteome remodeling during metastasis and serial passage of PDX tumors.

Non-murine proteins detected in at least 70% of samples were compared between all LUAD and LUSC samples (FDR < 0.001) (Fig. 4D). Supporting that our data captures accurate proteomic signatures of histology[34], markers of LUAD, TTF-1 (NKX2.1), Napsin A (NAPSA) and KRT7 have significantly higher expression in LUAD, whereas markers of LUSC, TP63, CK5/6 (KRT5/6 A) and SOX2 have significantly higher expression in

LUSC (Fig. 4E, F). Other significantly upregulated proteins that distinguished LUAD from LUSC (Supplementary Data 2), included MUC1, SLC34A2 and AGR2 for LUAD and KRT3, ADH7 and GBP7 for LUSC (Fig. 4E, F).

**LUAD and LUSC proteotypes associated with differences in patient survival and protein-phosphotyrosine signatures.** The subset of PDX human tumor proteome proteins identified in at least 70% of cases, was subjected to unsupervised consensus clustering. Among the 58 LUAD samples, which included two technical repeats, consensus clustering revealed 3 groups with high stability (Fig. 5A; Supplementary Fig. 3A–D). PCA of tumor/human proteome identified in all PDXs further supports the identification of three distinct proteotypes (Supplementary Fig. 3E), designated LUAD1, LUAD2 and LUAD3. Similar analysis with 62 LUSC samples identified 2 groups designated LUSC1 and LUSC2 (Fig. 5B; Supplementary Fig. 3F–J). Tumor clinical and genomic attributes were assessed for association with proteotypes. The proteotypes showed no association with sex or smoking (Fig. 5A, B). Although modest associations were observed among proteotypes and certain clinical and/or genomic features, altogether, these associations show that no single attribute can predict tumor proteotype (Supplementary Table 3).

Patient 5-year-OS was significantly different between LUAD1 and LUAD3 proteotypes, while LUAD1 and LUAD2 have similar OS (Fig. 5C; Supplementary Table 4). The 5-year OS of LUSC2 is significantly worse than LUSC1 (Fig. 5D; Supplementary Table 4). Stage, a known prognostic factor[35], shows a significant difference in 5-year-OS among LUAD but not LUSC (Supplementary Tables 5 and 6). Multivariate survival analysis with stage failed to show a significant survival difference between proteotypes, although the trend remains the same (Supplementary Tables 5 and 6). The loss of significance may be due to small sample size of different stages.

Deregulated protein tyrosine kinase signaling is a major hallmark of cancer and represents a major drug target class[36,37]. The subset of the proteome modified by tyrosine phosphorylation is a product of activated protein-tyrosine kinases and phosphatases, which are expressed at highly variable levels in cancer[38]. To profile protein pY in the PDX models, pY-peptides from trypsin-digested samples were purified by using an affinity enrichment method involving capture with immobilized Superbinder SH2 domain variants[39,40]. MS analysis of pY peptides showed a normal distribution based on numbers of pY peptides and maximum signals detected per sample (Supplementary Fig. 2H, I). Each LUAD and LUSC proteotype presented a distinctive pY profile, suggesting proteotypes differ in cell regulation processes controlled downstream of activated protein-tyrosine kinases and phosphatases (Fig. 5G, H; Supplementary Data 3). Notably, LUAD3 among LUADs and LUSC2 among LUSC, the proteotypes with the worst survival in each histological group, had the highest level of enriched pY sites (Fig. 5G, H).

**Proteotypes feature biological pathway vulnerabilities and candidate actionable targets.** Ingenuity pathway analysis (IPA) of significant differentially expressed proteins identified multiple enriched pathways for each proteotype (Supplementary Data 2). Top selected active pathways are shown in Fig. 6A, B. LUAD1 showed enrichment for many pathways including those involved in amino acid catabolism, aerobic metabolism, signaling by EGF, and TCA cycle (Fig. 6A). LUAD2 was associated with the fewest number of pathways, which included enhanced signaling of PTEN, BAG2, and apelin (Fig. 6A). LUAD3 was enriched for leukocyte extravasation and regulation of epithelial to mesenchymal transition (EMT) by growth factors, as well as signaling by integrins, Rho, and MAPK (Fig. 6A). LUSC1 was strongly enriched for spliceosome cycle,

pathways involved in translation, and DNA repair pathways (Fig. 6B). LUSC2 had enriched and active pathways such as VEGF and sonic hedgehog signaling (Fig. 6B).

Additional pathways characterized by proteotype-enriched pY-proteins, were identified (Supplementary Data 3), where the five most significant are shown in Fig. 6C, D. These included lower activity of integrin and leukocyte extravasation in LUAD1; increased activity of EGF and p70S6K in LUAD3 (Fig. 6C); and increased activity of cytoskeletal pathways such as actin cytoskeleton and ILK signaling in LUAD2 (Fig. 6C). Synaptogenesis and signaling pathways involving JAK family kinases were enriched for LUSC proteotypes (Fig. 6D; Supplementary Data 3).

In order to identify protein signatures that could be used to define the proteotype of primary tumors, we considered only significantly differentially expressed proteins as defined by having a more than 4-fold difference (≥4-fold, FDR < 0.05) between proteotypes, and detected in at least 50% of cases (Supplementary Fig. 4; Supplementary Data 2). This threshold was established based on published evidence that measurements of proteins with this magnitude of change were found reproducible and reliable, and with a high correlation rate between MS and western blot signals (Pearson's $r = 0.8–1$)[41]. For LUAD1 we identified 16 markers (Supplementary Fig. 4) including AGR2/3 and MUC5AC/B, which are highly expressed in mature secretory cells from the lung and gut[42] and NAPSA and SLC34A2, which are associated with surfactant metabolism in lung alveoli[43]. Therefore, most of the markers were associated with mature cell identity or mature cell function, particularly from the endoderm lineage. We identified 12 markers for LUAD3 (Supplementary Fig. 4), including DCBLD2, MCAM and VIM, which are considered EMT markers that are usually associated with tumor dedifferentiation and poor prognosis in NSCLC[44–46]. This is consistent with enrichment of the EMT pathway and worse OS in LUAD3 (Fig. 6A). Remarkably, in LUAD2 we identified 26 markers (Supplementary Fig. 4) including KRT5/6A, SERPINB3/4/5/13, DSG3, TRIM29 and LYPD3, which are related to squamous differentiation[47–49]. The expression of these markers was comparable to LUSC indicating a squamous-like aspect of the LUAD2 proteotype (Supplementary Fig. 4). Consistent with this observation, the majority of LUAD samples that clustered with LUSC comprise the LUAD2 proteotype (Fig. 4C). LUSC1 did not have many reliable markers that were expressed 4-fold higher than LUSC2 and hence the fold-change cut-off was reduced to 3 fold for this proteotype (Supplementary Fig. 4). Among these, SCGN, CHGA and CADM1 are known neuroendocrine tumor markers[50–52], suggesting this proteotype is neuroendocrine-like. Consistent with this observation, the majority of LUSC samples that clustered together with LCNEC are LUSC1 (Fig. 4C). LUSC2 showed high expression of 43 markers, many of which, including KRT6A/B/4/13, SERPINB3/4/13, DSG3, CSTA and GBP6 are related to squamous cell differentiation[47–49,53,54]. These findings suggest that LUSC2 is a proteotype of what is classically identified as LUSC, whereas LUSC1 may be a smaller group that is neuroendocrine-like.

To validate the proteotypes in unrelated cohorts we interrogated >100 LUAD primary tumor proteomes from Gillette et al. who identified 4 subtypes by multi-omic analysis[12], and >100 LUSC primary tumor proteomes from Stewart et al. who described two major groups *Redox* and *Inflamed*, and a minor *Mixed* group[14]. Our defined proteotype signature proteins, identified in at least 70% of the cohort, were used for unsupervised hierarchical clustering (Supplementary Fig. 5A, B). Interestingly, this stratified the LUAD and LUSC patients into groups corresponding to our proteotypes (Supplementary Fig. 5A–D), and with considerable overlap with the subtypes identified in the original studies (Supplementary Table 7).

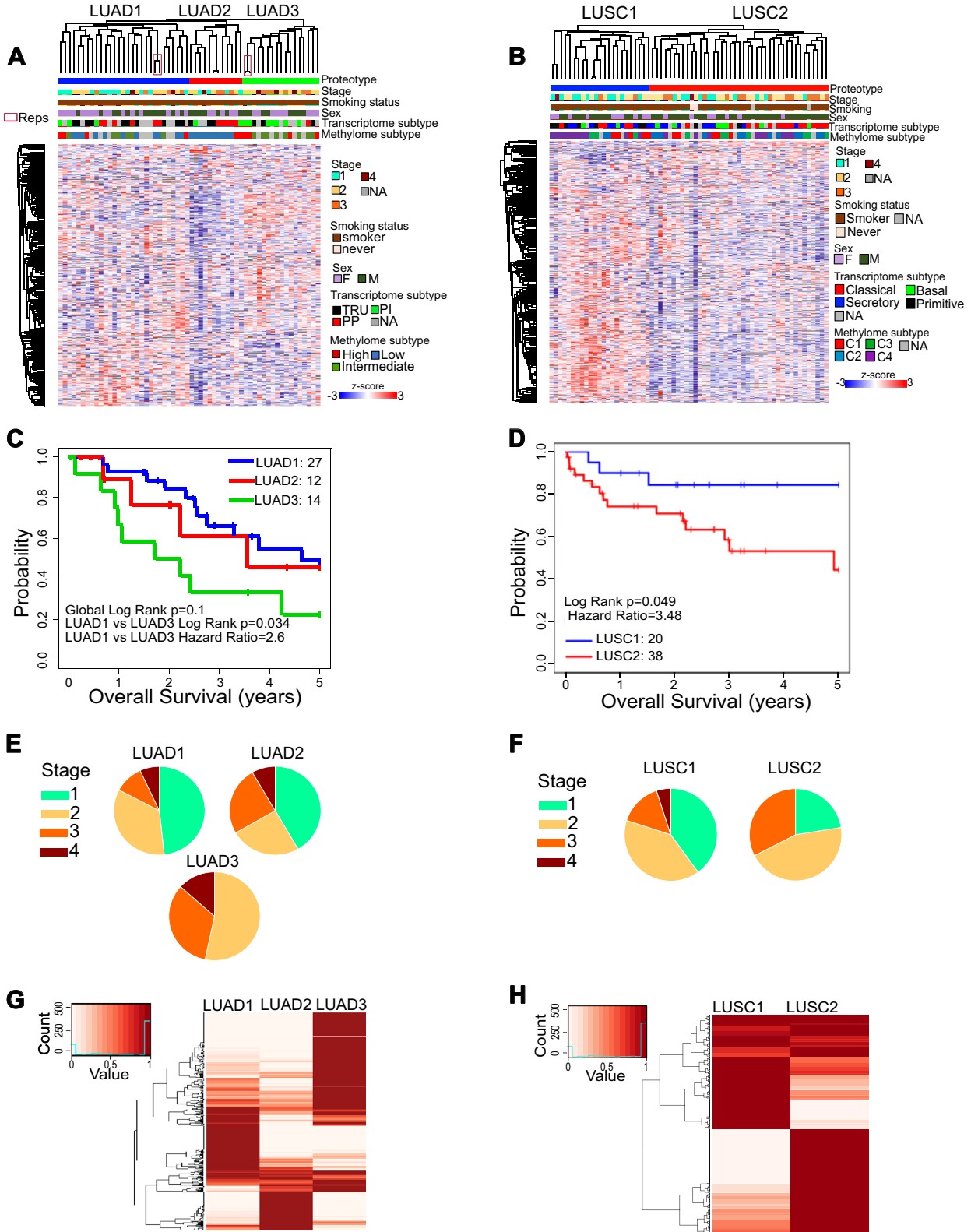

**Fig. 5 Lung LUAD and LUSC defined proteotypes associated with differences in patient survival and protein-phosphotyrosine signatures. A**, **B** Tumor proteome separates (**A**) LUAD histology PDX samples into three distinct subtypes LUAD1, LUAD2, and LUAD3 and **B** LUSC into two distinct subtypes LUSC1 and LUSC2. See also Supplementary Fig. 3A–J. **C**, **D** 5-year overall survival Kaplan–Meier plot shows significant survival differences between the **C** LUAD1 and LUAD3 groups and **D** the two LUSC groups. See also Supplementary Table 4. **E**, **F** Stage breakdown of **E** LUAD and **F** LUSC proteotypes. **G**, **H** Proteotypes of **G** LUAD and **H** LUSC have distinctive protein-pY profiles.

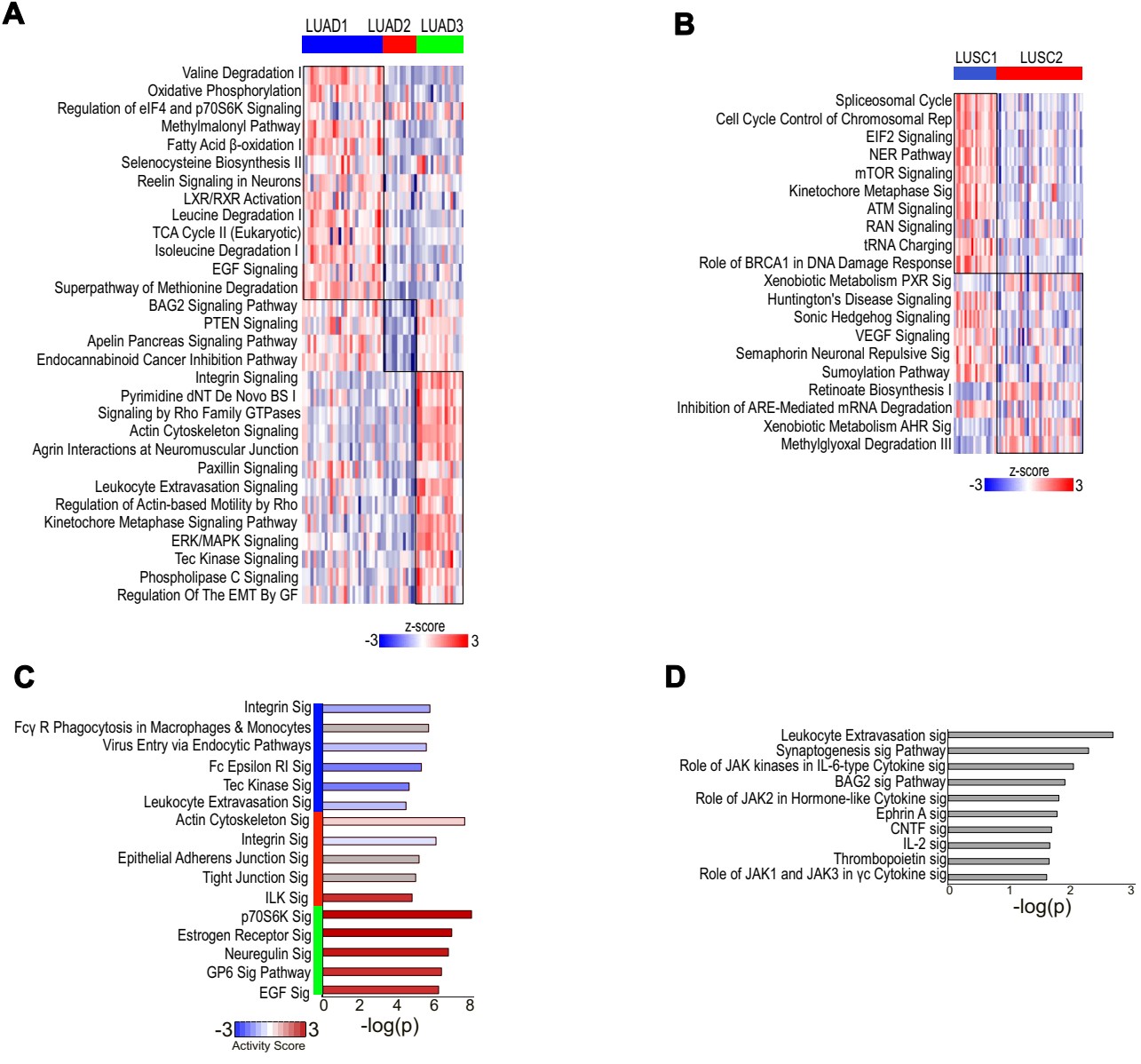

**Fig. 6 Proteotypes feature active biological pathway. A**, **B** Enriched and active pathways of each **A** LUAD and **B** LUSC proteotypes by total proteome is shown. The heatmap shows gene set score of the pathway for each sample. See also Supplementary Data 2. **C**, **D** Top significant pathways of **C** LUAD and **D** LUSC proteotypes by pY proteome is shown. See also Supplementary Data 3.

The proteotype protein signatures were also able to effectively categorize 34/34 LUAD and 9/12 LUSC cell lines previously characterized at the proteome level as part of the DepMap project[27]. The LUAD cell lines clustered into groups corresponding to LUAD1 (8 lines), LUAD2 (13 lines), and LUAD3 (13) (Supplementary Fig. 6A, B), and the LUSC cell lines clustered into groups corresponding to LUSC1 (6 lines) and LUSC2 (3 lines) (Supplementary Fig. 6C, D). Genetic and pharmacological sensitivities associated with the cell lines grouped by proteotype were defined by using DepMap data[28] (Supplementary Data 4). This analysis revealed proteotype-specific sensitivities (Supplementary Data 4), including a top set of candidate actionable targets and molecules based on the effect size of their inhibition on cell line viability (Supplementary Fig. 6E, F). We further identified sensitivities that matched significantly differential proteins of the proteotypes (Supplementary Data 4). Pathway analysis of these matched targets showed the LUAD1 lines to be sensitive to losing components of the TCA cycle,

LUSC1 spliceosome and LUSC2 ribosome biogenesis. This was consistent with the enrichment of high activity of TCA cycle in LUAD1 and higher enrichment and activity of spliceosome in LUSC1 (Supplementary Data 4). Ribosome components were expressed lower in LUSC2, which might be why LUSC2 cell lines are sensitive to losing components of ribosome biogenesis.

**Proteotypes demonstrate recurrent genomic alterations**. To assess proteotype-associated genome alterations the frequencies of mutation and CNVs were assessed (Fig. 7A, B) (Supplementary Data 1). LUAD1 enriched for alterations in *EGFR, MN1* and *MYH9*, LUAD2 enriched for *BRCA2*, while LUAD3 did not enrich for known cancer drivers. LUSC1 enriched for alterations in *DUSP22, KSR2, ATXN2* and *RAD50* and LUSC2 enriched for *KEAP1*. The enrichment of these cancer driver alterations in proteotypes might explain some of the differences seen at the proteome level. For instance, enrichment of *EGFR* alterations in

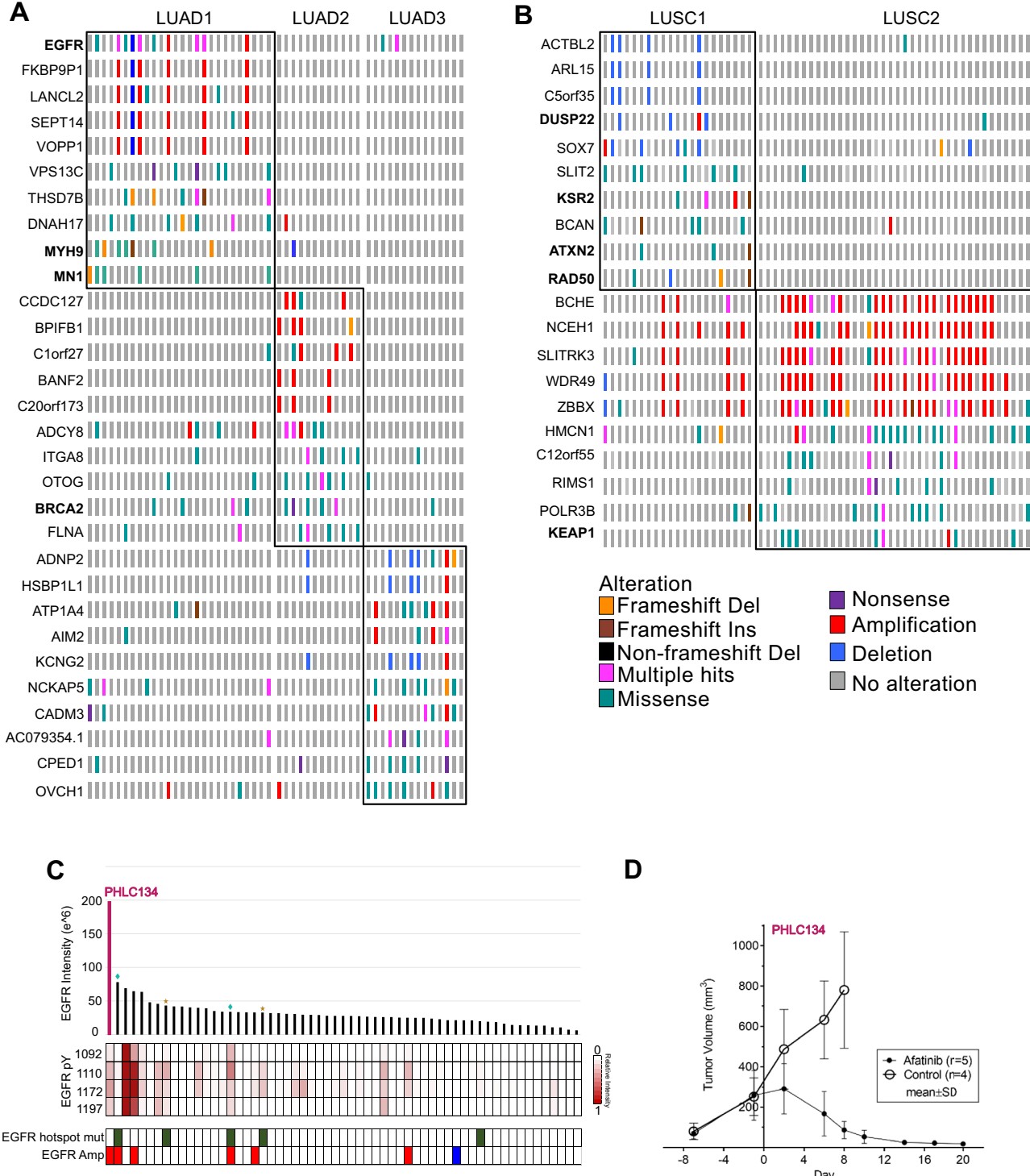

**Fig. 7 Proteotypes demonstrate recurrent genomic alterations. A**, **B** Significantly altered genes associated with **A** LUAD and **B** LUSC proteotype are shown. Genes in bold are cancer drivers as defined by cBioportal. **C** EGFR protein expression, active pY sites, driver mutation and amplification status per LUAD cases. Blue diamonds show responders and yellow star cases show non-responders to EGFR TKI. **D** Tumor volume growth trend in response to Afatinib ($n = 6$) vs. Control ($n = 5$) in NSCLC PDX model 134 that has amplification and elevated protein expression of WT EGFR (mean ± SD) (linear mixed effects model, $p$-value = 2.3E-15). Source data are provided as a Source Data file.

LUAD1 is consistent with the activation of EGF signaling pathway noted for LUAD1 (Figs. 6A and 7A). Interestingly, subtypes enriched in EGFR alterations in other studies[31,55] and LUAD1 herein are subtypes with relatively better clinical outcomes. EGFR tyrosine kinase inhibitors (TKIs) are currently restricted to cases with EGFR hotspot mutations. We observed that cases with

hotspot mutations do not always have high EGFR expression nor increased activated pY sites compared to cases without mutations (Fig. 6C). Herein, we demonstrate that a PDX with WT amplification and high EGFR protein expression and pY enrichment but no oncogenic hot spot mutation significantly responded to treatment with the EGFR TKI Afatinib (p-value = 2.3E-15)

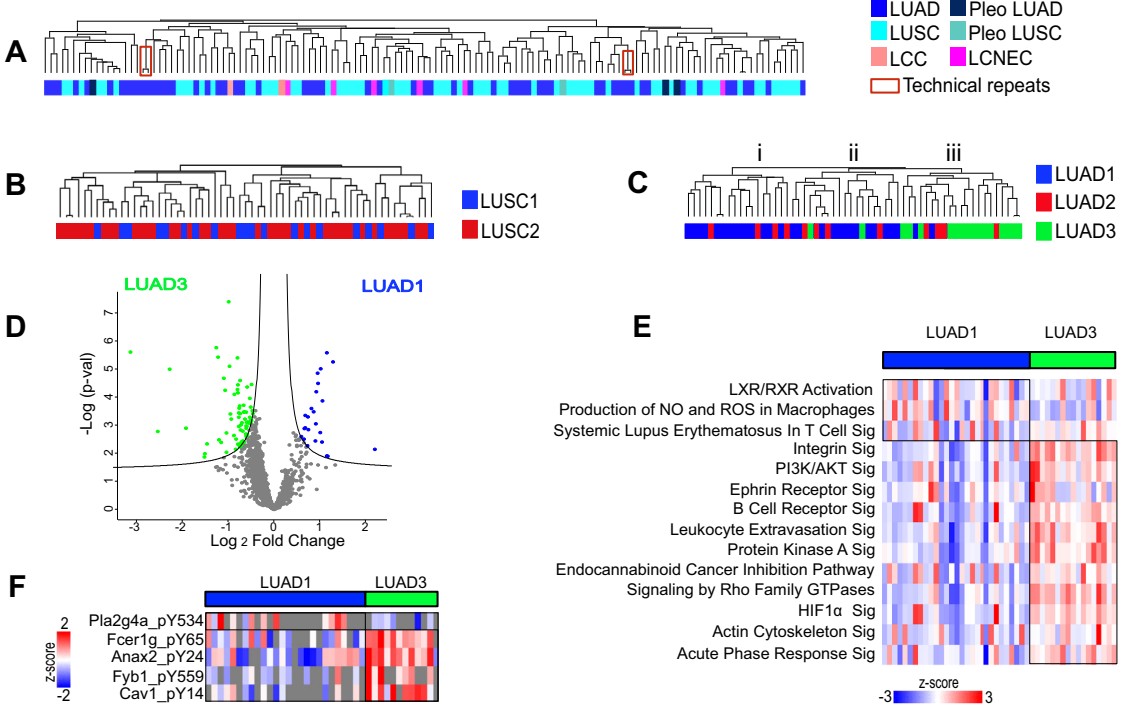

**Fig. 8 Differential stromal composition of LUAD proteotypes 1 and 3. A–C** Unsupervised hierarchical clustering of stromal (mouse) proteome does not cluster based on **A** histological type nor **B** LUSC proteotypes. **C** LUAD3 is significantly enriched in stromal proteome cluster (iii) (Fisher's exact *t*-test *p*-value = 3.7E-07). **D** Significantly differential stromal proteins between LUAD1 and LUAD3 (two-sided *t*-test, FDR < 0.05). See also Supplementary Data 2. **E** Significant and active pathways that differ between LUAD1 and LUAD3, based on proteins identified in Fig. 7D. The heatmap shows the gene set score of the pathway for each sample. See also Supplementary Data 2. **F** Significantly differential pY peptides between LUAD1 and LUAD3 (two-sided *t*-test, *p*-value < 0.05).

(Fig. 7E). This is an example wherein proteome analysis might reveal potentially responsive, activated target pathways not uncovered by genome/transcriptome analyses. In a previous study, we treated 4 PDX models bearing EGFR activating mutations by multiple EGFR inhibitors[56]. Two models (PHLC137 and 192) (Fig. 7C-indicated by blue diamond) responded to Erlotinib, Dacomitinib, Afatinib and Cetuximab, whereas the other two models either did not, or only responded to cetuximab (PHLC148 and 164) (Fig. 7C-indicated by yellow star). Comparing the proteome of the two responders vs. non-responders revealed VIM expression significantly higher in the non-responders, consistent with previous reports[57], and CALML3 expression higher in responders (Supplementary Fig. 7A). Another potential TKI target in NSCLC is fibroblast growth factor receptor 1 (FGFR1), which is amplified in 20% of LUSC and known to be involved in cell proliferation and survival. Indeed, 24% (14 of 58) of our LUSC cases contain an amplification of 8p11.23 that includes FGFR1(Supplementary Fig. 7B). However, we saw no significant change relative to non-amplified samples in the levels of proteins or protein-pY in these cases (Supplementary Fig. 7C). Consistent with these observations, we tested the FGFR1 inhibitor BGJ398 for growth inhibition in four randomly selected FGFR1-amplified tumors (PHLC-200, −274, −299, and −321), and found that in all cases there was an initial minor shrinkage of tumors but ultimately no inhibition of PDX tumor growth (Supplementary Fig. 7D–G). These examples demonstrate the utility of the PDX models to test therapeutic hypotheses including target validation based on protein expression or pathway activation.

**Differential stromal composition of LUAD proteotypes 1 and 3.** Unsupervised hierarchical clustering of the stromal proteome, which is of murine origin, and including proteins identified in at least 70% of samples did not distinguish histological subtype (Fig. 8A) or proteotype in LUSC (Fig. 8B). Interestingly, LUAD1 and LUAD3 had distinctive stromal proteomes (Fig. 8C), with LUAD3 significantly associating with cluster iii (Fisher's exact *t*-test *p*-value = 3.6E-07) and LUAD1 significantly associating with cluster i and ii (Fisher's exact *t*-test *p*-value = 4.9E-06). LUAD2 did not significantly correlate with either of these clusters (Fisher's exact *t*-test *p*-value > 0.05), suggesting it does not establish or maintain a distinct stromal composition. Stromal proteins significantly differentially expressed between LUAD1 and LUAD3 were identified (Fig. 8D; Supplementary Data 2). The active pathways enriched by significantly differential stromal proteome are shown in Fig. 8E. LUAD1 was enriched for pro-inflammatory pathways such as systemic lupus erythematosus in T-cell signaling, and nitrogen oxide and reactive oxygen species production of macrophages (Fig. 8E). LUAD3 was enriched for signaling of acute phase response, leukocyte extravasation and B cell receptor, and pathways involved in EMT such as signaling of integrin, actin cytoskeleton, protein kinase A and Rho. Murine (i.e., stromal) pY peptides were also compared between LUAD1 and LUAD3. Tyrosine phosphorylation of cytosolic phospholipase A2 (pla2g4a), a key enzyme for generation of pro-inflammatory eicosanoids was significantly higher in LUAD1 (Fig. 8H). Fcer1g, Anax2, Fyb1 and Cav1 were significantly higher in LUAD3, where Cav1-pY14 is known to facilitate metastasis[58], Fyb1-pY559 is known to alter T-cell adhesion[59] and Fcer1g is known to regulate immune homeostasis[60]. Two of the top identified candidate targetable molecules of LUAD3 are LGALS1 and NT5E (Supplementary Table 9), which have been associated with immune evasion[61,62]. These observations suggest that signaling from tumor cells involving proteins such as NT5E and LGALS1 in LUAD3 might influence remodeling of the stromal microenvironment.

## Discussion

Our PDX resource included 137 models and provides insights into lung cancer including the stratification of the major histological subtypes of LUAD and LUSC into proteotypes with prognostic impact and associated with distinctive tumor and in some instances stromal signatures of activated and targetable pathways and processes. The PDX models, which represent particularly aggressive tumors[15], were comprehensively profiled at several molecular levels including gene expression, gene copy number variation, DNA methylation, exome mutations, and tumor cell and stromal proteomes and pY-proteomes. Interrogation of these datasets including comparison with primary tumor profiles validated them as models of NSCLC. DNA alterations linked to NSCLC outcome were represented in the PDX models such as enrichment of *KRAS* mutations associated with worse outcome, and bias against *EGFR* mutation associated with better outcome. LUSC methylation subtype C1 was underrepresented where C2 was over-represented. Transcriptome-based subtypes defined by analysis of primary NSCLC were also present in the PDX models. Nevertheless, the PDX models will have utility for the testing of hypotheses, for example related to therapeutic vulnerabilities based on their genome, transcriptome and methylation signatures that represent primary tumors.

The sequence-to-phenotype continuum Genome→Transcriptome→Proteome→Cancer implies that the cancer phenotype is largely a product of the proteome. However, the analysis of the PDX models, consistent with findings made with primary tumors, indicates once again that cancer proteome remodeling is largely unpredictable based on measures of gene or transcript. The three LUAD and two LUSC proteotypes are associated with significant differences in candidate biomarkers, protein-pY modifications, activated signaling pathways and cellular processes, and patient survival. Interestingly, proteotype LUAD2 was found to be squamous-like, while proteotype LUSC1 appears neuroendocrine-like. EGFR TKIs are currently one of few targeted therapies for NSCLC[4,63] and with a requisite for EGFR activating mutations[64,65]. Amplification and protein expression of WT *EGFR* are not consistent biomarkers for anti-EGFR responsiveness[66–71], and immunohistochemistry analysis of EGFR protein can be unreliable[72]. Furthermore, we found that a WT EGFR model (PHLC 134) was sensitive to the EGFR TKI Afatinib. These observations, albeit made with a small sample set illustrate the utility of the PDX models to investigate tumor biology and therapeutic hypotheses, and the potential for proteotype classification as a predictor of response.

The importance of stroma in tumor development and phenotypes including response to treatment has been discussed extensively. NSCLC-associated fibroblasts have been linked to resistance to EGFR TKIs, and fibroblast activation protein-positive fibroblasts have been implicated in immunotherapy resistance[73,74]. Characterizing the stromal composition of a tumor could therefore aid in clinical actions. The MS-based proteome platform analyzes samples that have been digested into peptides. Consequently, those peptides that differ between mouse and human can be used to support the conclusion that the cognate protein from that species has been identified. However, given the high degree of sequence identity between the two species, shared peptides are frequently seen that cannot be distinguished as mouse or human. In these instances, based on common strategies used in the field, signals from shared peptides may be attributed to the protein with the greater number of species-specific peptide identifications. This approach supports determination of mouse/stroma and human/tumor content in PDX samples. However, for any given protein, complementary analyses such as immunocytochemistry and targeted, quantitative MS may further inform on localization and relative abundance.

By leveraging the cross-species nature of the PDX proteome, we were able to correct for discrepancies in tumor/stromal composition, which has not been possible with primary tumor analyses. We did not discern significant differences between LUAD and LUSC stromal proteomes, consistent with studies that demonstrated almost no differences in their respective stromal transcriptomes[75]. However, proteotypes LUAD1 and LUAD3 were found to have discernibly different stromal proteomes. This suggests the proteotype-specific molecular signatures can recruit distinct stromal compartments to the microenvironment. Similar phenomena has been reported in breast cancer, where subtypes identified among breast cancer PDXs using the tumor-specific proteome can also be distinguished by stromal proteomes[76]. Leukocyte extravasation, which is the general recruitment of different stromal components such as monocytes, neutrophils and lymphocytes was significantly enriched both at the total proteome and pY-protein level in LUAD3, suggesting a higher recruitment of leukocytes in this most aggressive proteotype. An obvious drawback of our method was the use of subcutaneous implantation of patient tumors in NOD-SCID mice that have a compromised immune system. NOD-SCID mice have a reduced innate immunity and nearly no adaptive immunity[77,78]. Although, these immunodeficient features are necessary to prevent tumor rejection, this limits the scope of stromal differences that would usually be present in an immune-healthy individual. This also makes NOD-SCIDs not the ideal model for testing of immuno-modulatory treatments. In LUAD patients, immune "hot" and "cold" subtypes were recently described[12], where the hot subtype was identified by their stronger signature for B and T cells and macrophages, while also presenting stronger signatures for immune inhibitory cells and processes[12]. We observed that the hot subtype corresponds to the LUAD3 proteotype. Another obvious limitation of the PDX models is that they are not readily useful for comprehensive/systems type analyses of gene dependencies or chemical screens.

The NSCLC PDX models, multi-omics datasets, and the recognition of defined proteotypes represent invaluable resources that may assist in the exploration of NSCLC biology and the pre-clinical development of new treatments. We identified several proteotype-associated candidate targets and drugs not conventionally used in the clinic for treatment of NSCLC, although some have been or are currently in clinical trials. This portends the classification and treatment of cancers according to proteotype and guided by pre-clinical studies with patient-derived model systems.

## Data processing

**Whole-exome sequencing quality control preprocessing**. Xenome (v1.0.1)[79] was used to filter out mouse stromal reads by aligning the reads to DNA of the NOD-SCID mouse., the remaining reads were aligned to the human reference genome (hg19) using Burrows-Wheeler Aligner (v0.7.12)[80]. Quality control, local realignment of indel, base quality score recalibration (BQSR), duplicate reads marking and further processing of the mapped reads were performed using the standard Genome Analysis Toolkit (GATK) pipeline (v3.4)[81], samtools (v1.2)[82] and Picard (v1.140) (https://broadinstitute.github.io/picard/). The pipeline generated a single Binary Alignment Map (BAM) file for each sample (either PDX or matched normal) that includes reads, calibrated quantities, and alignments to the genome.

**DNA methylation quality control preprocessing**. The R packages of minfi (v1.6.0)[83] and Illumina Human Methylation 450k manifest (v0.4.0) were used to processed the idat files. The data was background corrected and normalized using the ssNoob

| Table 1 The possible scenarios for peptide assignment to Human or Mouse. | | |
|---|---|---|
| **Human unique peptides** | **Mouse unique peptides** | **How razor peptides are assigned to a protein group:** |
| Yes | Yes | Unique peptides used to report a quantification value for each human and mouse protein. Razor peptides contribute only to the protein group with largest number of peptide identifications. |
| Yes | No | Unique and Razor peptides all used to report a value for Human protein group; no mouse reported. |
| No | Yes | Unique and Razor peptides all used to report a value for Mouse protein group; no human reported. |
| No | No | All razor peptides are used to report a Human/Mouse value for protein |

method. The batch effects were assessed using the Source of Variability and Principal Components approach in the sva package. Quality control was performed to identify failed samples (detection p-value > 0.05). The following probes were filtered out prior to data analysis: 58,771 failed probes (detection p-value > 0.01); 43,250 mouse-related probes for potential contamination from mouse; 8,440 probes on the x,y sex chromosomes to avoid sex-related methylation biases; 12,128 probes with single-nucleotide polymorphisms (SNPs) at CpG sites; and 22,517 cross-reactive probes. DNA methylation scores for each CpG site were calculated as a beta ($\beta$) value ($\beta = (M/(M + U))$) in which M and U indicate the mean methylated and unmethylated signal intensities for each assayed CpG or CpH, respectively.

**Gene expression data quantification**. The data files were quantified in GenomeStudio (Illumina). All the samples passed Illumina sample dependent and independent QC Metrics. The data files were loaded into the R package of lumi (v2.24.0)[84]. Data from the individual platforms were $\log_2$ transformed, background corrected, normalized using quantile normalization and quality control. Batch effects were adjusted using ComBat, implemented in the sva package (v3.20.0).

**Copy number alterations data quantification**. The data files were quantified in GenomeStudio (Illumina), normalized against controls, background subtracted, and GC wave corrected. The signal intensity as Log2 R ratio (LRR) and B allele frequency (BAF) values were used to derive copy number, purity and ploidy of PDX with matched normal using ASCAT(v2.5, allele-specific copy number analysis of tumors)[85].

**Database searching and quantification of total proteome**. The acquired raw data was searched against reviewed Uniprot Human and Mouse Reference databases proteome using MaxQuant search engine. For both search algorithms, the parent and fragment mass tolerances were set to 20 ppm and 0.5 Da, respectively. Only complete tryptic peptides with a maximum of two missed cleavages were accepted. Sum of unique+razor peptides were used for quantitation, with a minimum of 1 razor+unique peptide required. Methionine oxidation and protein N-terminal acetylation were included as variable modifications, while carbamidomethylation was considered fixed modifications. Search results were filtered using a strict false discovery rate (FDR) of 0.01. TMT reporter ions were quantified using MaxQuant reporter ions quantifier node with an integration tolerance of 20 ppm, on the MS order of MS3.

There is a high degree of sequence redundancy in the proteome. In bottom-up proteomics, this leads to situations where often peptides cannot be uniquely associated with one protein of origin. This issue is further complicated in mixed species PDX samples where some shared peptide sequences are identical in human and mouse. In MaxQuant's 'Unique+Razor' strategy, this complexity is addressed by using unique peptides to

form distinct protein groups (i.e., human only or mouse only) and with razor (i.e., shared) peptides contributing only to the protein group with the greater number of peptide identifications[86] (Table 1). In situations where there are no unique peptides for a protein, the shared peptides are still used to form a protein group but since the specie-of-origin is ambiguous, they are designated as Human/Mouse (Table 1).

**Total proteome data normalization**
*Intra-TMT experiment group normalization.* Samples were normalized to the sample with the maximum sum intensity of each TMT experiment group. Briefly, sum intensity of all protein for each sample was measured. The sample with the maximum intensity in each TMT group was identified. A conversion factor is calculated which is then multiplied by all proteins of that sample.

*Inter-TMT experiment group normalization.* To normalize TMT groups to each other, internal reference scaling method was used as previously described[87]. Briefly, the control channels containing the pool of tumors (channel 126 and 131 of each batch) within each TMT experiment were averaged and used to create reference values per protein per each batch. The reference values for each protein in each TMT group were then averaged (geometric mean), and scaling factors calculated for each protein to adjust its reference value to the geometric mean value was measured. These scaling factors were then used to adjust the summed reporter ion intensities for each protein in the remaining eight experimental samples in each TMT experiment.

*For tumor/stroma content normalization.* To normalize tumor/stroma content, two conversion factors, one for human-specific proteins and one for murine-specific proteins, were calculated for each sample. To calculate each conversion factor, the sum of total intensities of human-specific proteins for all samples was divided by the number of samples. Then, for each sample, this average value was divided by the sum of total intensities of human-specific proteins for that sample, yielding its conversion factor. The same strategy was employed to calculate a conversion factor for mouse-specific proteins. Normalized protein group values were calculated as the product of measured intensities times the sample-specifc conversion factor. Refer to Supplementary Data 1 for normalized values.

**Database searching and quantification of phosphoproteome**. Raw MS data were searched with MaxQuant on human and mouse database without match between runs and default parameters were used for peptide and protein search. FDR for protein and peptide selection is less than 0.01. Phosphorylation of serine/threonine/tyrosine is used for variable modifications. pY site localization probability is higher than 0.75. MS data of CID and HCD from the same sample were combined as different fractions.

## Data analysis

### Genomic data analysis

*Somatic mutation Calling.* For the somatic mutations calling, single-nucleotide variants (SNV) calling were using MuTect (v1.0)[88] and indels were using VarScan (v2.3.8)[89]. ANNOVAR[90] were used to annotate all the mutations. For samples without a matched normal, additional filtering was done by using public databases from dbSNP (version 138 flagged), ExAC03 and ESP6500. Tumor Mutation burden (TMB) of a PDX sample was calculated by the number of non-synonymous somatic mutations per mega-base in coding regions. The Oncoprint plots were generated using significant mutated genes in lung adenocarcinoma or squamous cell cancers reported from Campbell et al. study[32].

*Copy number alterations analysis.* Significant regions of aberrations were identified using GISTIC(v.2.0.23)[91]. The gene-level copy numbers were also obtained from GISTIC, and the gene was considered as copy gain or loss if the gene-level CNV value were larger than 0.3 or smaller than −0.3, respectively.

*DNA methylation hierarchical clustering.* Unsupervised clustering using the top 4,000 most variable CpGs (in promoter region or within 1500 bases of the transcription start site) defined by the standard deviation was carried out using Hierarchical clustering with Euclidean distance and linkage as ward.D2 in hclust of R package.

*DNA methylation 3D PCA.* Unsupervised methylation analysis was performed with dimensionality reduction approach and PCA was performed after probe exclusion (according to methodology) to understand the data with different tumor subtypes (Supplementary Fig. 1C) and identify sample outliers using the Partek Genomics Suite software (Partek, St. Louis USA).

*Expression subtype detection.* Wilkerson et al. suggested and validated expression subtypes of lung adenocarcinoma (*Terminal Respiratory Unit, Proximal-Proliferative* and *Proximal-Inflammatory*) and lung squamous (*Basal, Classical, Primitive* and *Secretory*) based on the gene expression characteristics[6,7,31]. These gene expression subtypes are highly robust across different cohorts and expression profile platforms. In our analysis, we used the Wilkerson et al. public subtype predictor centroids to predict the subtypes with a nearest centroid. The Pearson correlations were calculated between the predictor centroids and PDXs or TCGA using the centroids' genes (505 genes for lung adenocarcinoma, 208 genes for lung squamous). The subtype prediction was given by the centroid with the largest correlation value. We also calculated the frequencies of the expression subtypes in PDX models or TCGA. The subtype expression patterns were highly concordant between our PDX models and TCGA (Supplementary Fig. 1A, B).

### Proteomic analysis

*Proteotype assignment using consensus clustering.* Consensus clustering was performed using subset of tumor proteome quantified in at least 70% of the samples for LUAD and LUSC samples separately using *ConsensusClusterPlus*. Pearson correlation for the distance metric and Average method for the linkage algorithm, with 1000 resampling steps (epsilon = 0.8) was used (Supplementary Fig. 3A, F). The optimal number of clusters was identified using proportion of ambiguously clustered pairs method, where the cumulative density frequency plot exhibits a flat middle portion for the true number of k[92] and clusters are at least consisted of 3 samples (Supplementary Fig. 3B, C, G, H).

This method led to identification of 3 and 2 clusters for LUAD and LUSC respectively, where each cluster had >0.8 cluster consensus score indicating high stability of proteotypes (Supplementary Fig. 3D, I)[93].

*Proteome hierarchical clustering.* Proteome hierarchical clustering was performed by using the subset of tumor (human) proteome quantified in at least 70% of samples in the clustering (Figs. 4C and 5A, B) and using the subset of stromal (mouse) proteome quantified in at least 70% of samples in the clustering (Fig. 7A–C). Protein expressions are log2 transformed and z-score across respective samples. For sample (column-wise) and protein (row-wise) clustering, Pearson correlation distance with average linkage was used using Perseus software default parameters.

*Principal component and volcano analysis.* Only a subset of tumor proteome quantified in all samples were used for principal component analysis. Only the two components with the highest proportion of variance were picked for plotting the PCA plot using Perseus software. Volcano analysis was performed for proteins with detection in at least 70% of samples with indicated FDR cut offs and s0 = 0.1 using Perseus software.

*Analysis of differentially expressed proteome and pathway.* Permutation-based FDR corrected two-tailed student's *t*-test (*q*-value < 0.05) between one subtype compared to others in that histology type was performed on the entire proteome (human, mouse and ambiguous) (Supplementary Data 2). These proteins along with associated experimental expression and q-values were inputted in ingenuity pathway analysis[94]. Resulting enriched pathways (Supplementary Data 2) were further filtered based on significance p-value < 0.05 and activity score as determined by IPA z-score > or <0 (Figs. 6A, B and 8E).

*Gene set score analysis.* The gene set score of each pathway was calculated by averaging the z-score value of all proteins that enriched for the pathway for that sample, these values could then be presented in heatmap format to show the expression pathway per sample (Fig. 6A, B and 8E).

*Phosphoproteome and pathway.* A total of 564 and 484 pY sites were quantified in LUAD and LUSC samples, respectively. Tyrosine phosphorylation was analyzed in a supervised manner for each proteotype. pY signals for each pY site were divided by the maximum signal measured for that site to present the values in a relative manner compared to the maximum value of 1. Then, the average of relative values for each pY site was used for supervised clustering according to proteotypes. Phosphopeptides significantly different by two-tailed student *t*-test (*p*-value < 0.05) in one proteotype compared to the others were determined, and used for Ingenuity Pathway Analysis (Supplementary Data 3)[94].

## Methods

The University Health Network (UHN) Human Research Ethics (09-0510-T) and Animal Care (AUP603 for model establishment and AUP743 for drug study) Committees approved this study protocol. Animal care followed the guidelines of UHN Research Institutes' policies and the guidelines of the Canadian Council on Animal Care, and consistent with ARRIVE guidelines for study design.

**Human subjects**. A total of 500 patients were included in this study. The number of NSCLC cases was determined by specimens that had excess tissues for research between 2005 and 2014. Tumor specimens were collected at the Toronto General Hospital (TGH-UHN) with informed consent from participants using The University Health Network (UHN) Human Research Ethics protocol 09-0510-T. Human research followed the guidelines of Canada Tri-Council Policy Statement, in accordance with Declaration of Helsinki (www.pre.ethics.gc.ca.). No participant compensation was provided (Supplementary Tables 1, 2, 8).

**Clinical data annotation**. Clinical data can be accessed and downloaded from cBioPortal. The demographics, histopathological, stage and relevant clinical information is summarized in Supplementary Table 2 and Supplementary Data 1.

**Generation of PDXs in NOD-SCID mice**. As previously described[26], patient tumor samples were divided into 2 mm pieces, mixed with 4 °C 10% Matrigel and implanted into the subcutaneous flank tissue of male, age 4–6 weeks NOD.CB17-Prkdcscid (NOD-SCID) mice. Mouse replicates in PDX studies were stratified randomly to each group, to equally distribute tumor volumes and mouse body weights. Tumor volume was calculated ($V = W^2 \times L/2$) from twice weekly measurements for the length (L, largest diameter) and its perpendicular width (W), including skin fold using a caliper. A humane endpoint was reached at tumor size 1.5 cm in largest diameter; this was not exceeded in this study. Mice were euthanized and tumors are implanted/passaged serially into new NOD-SCID mice once they reach 1–1.5 cm in diameter, as measured with calipers. This process was repeated at least three times to establish a stable PDX. Harvested tumors were divided equally for cryopreservation, quick freeze in liquid nitrogen and preparation of formalin-fixed paraffin-embedded tissue blocks. Comprehensive profiling of PDXs was conducted from early passages 2–5.

**PDX drug screens**. PDX drug screens were performed using banked cryopreserved PDX fragments that were thawed and implanted in donor male, age 4–6 weeks NOD-SCID mice. (at passages <10), tumors were harvested at full size, and fragments were generated and implanted in male mouse replicates for drug treatments. When tumor volume replicates averaged 200 mm3, mice were randomized ($n = 6$/group) for treatment with vehicle or anticancer agents. The pan-FGFR1 tyrosine kinase inhibitor (TKI) BGJ398 was administered at 25 mg/kg/day, oral gavage, in a suspension in PEG300/D5W (2:1, v/v) (0.5% hydroxyethyl-cellulose, 0.2% Tween 80, 99.3% distilled water). Afatinib is an irreversible EGFR/ErbB2/ErbB4 tyrosine kinase inhibitor administered at 25 mg/kg/day, oral gavage, as a suspension in 0.5% (w/v) methylcellulose suspension-0.4% Tween 80. Tumor sizes and mouse body weights were measured twice weekly. Research grade drugs were purchased from UHN-Shanghai Research & Development Co., Ltd (Shanghai, China).

**Whole-exome sequencing sample preparation and data acquisition**. PDXs profiled for whole-exome sequencing included 122 models. For all DNA-based profiling, DNA was isolated using a gSYNC™ DNA Extraction Kit (FroggaBio Cat# GS100) following user guide directions. In brief, flash frozen tumor up to 25 mg of fragments were dissociated with 200 µL of GST Buffer and 20 µL of Proteinase K then vortex thoroughly and incubated at 60 °C overnight. This extraction method was based on using a DNA spin column with buffers and centrifugation to remove impurities, and finally eluting the purified DNA. The exome capture was performed using the Agilent SureSelect Human All Exon 50 Mb kit (Agilent Technologies, Santa Clara, CA) according to the manufacturer's instructions. The captured DNA of PDX models and their matched normal were sequenced on the Illumina HiSeq 2000 platform (Illumina, San Diego,CA), and paired-end sequences (2 × 101 bp) were generated for each sample.

**DNA methylation sample preparation and data acquisition**. PDXs profiled for DNA methylation included 102 models. The Illumina Infinium HumanMethylation450k BeadChip array (Illumina, San Diego, CA, USA), which includes 485,512 CpG sites, was used for the interrogation of methylation profiles of PDX models. Genomic DNA for each PDX was treated with sodium bisulfite using EZ DNA Methylation Kit (Zymo, Irvine, CA). The bisulfite-converted DNA sample were amplified, enzymatic fragmented and processed as specified by the manufacturer's instructions. The hybridized BeadArrays are scanned by the Illumina iScan array scanner and then, raw intensities data (.idat) of both cy3 and cy5 were obtained after scanning.

**Gene expression sample preparation and data acquisition**. PDXs profiled for gene expression included 92 models. RNA as directed by user guide, RNA was isolated from liquid nitrogen flash frozen PDX tumor fragments using TRIzol RNA Isolation Reagents (Thermo Fisher Cat# 15596026). Fragments were lysed and homogenized in TRIzol™ Reagent using 1 mL of TRIzol™ Reagent per 50–100 mg of tissue and incubated for 5 min to allow complete dissociation of the nucleoproteins complex. Then mixed with chloroform, 0.2 mL per 1 mL of TRIzol™ Reagent, and incubated and centrifuged for phase separation. The aqueous phase containing RNA was transferred to a new tube with 0.5 mL of isopropanol per 1 mL of TRIzol™ reagent used for lysis. The pellet was collected after centrifugation and resuspended in 1 mL of 75% ethanol per 1 mL of TRIzol™ reagent used for lysis and pelleted and resuspended in 20–50 µL of RNase-free water. RNA cleanup was performed using a RNeasy Mini Kit (Qiagen Cat No. 74104) as directed in user guide. RNA of PDXs was labeled and amplified using the human-specific Illumina Whole-Genome Gene Expression Direct Hybridization or Whole-Genome DASL assay kit according to the manufacturer's protocol (Illumina). The labeled samples were hybridized to HumanHT-12 v4.0 Gene Expression BeadChips, incubated at 58 °C for hybridization for 18 h. The BeadChips were then washed and stained as per the Illumina protocol and scanned on the iScan (Illumina).

**Copy number analysis sample preparation and data acquisition**. PDXs profiled for copy number variation included 112 models. DNA was extracted as described above from each PDX or germline sample (their matched patient normal) was hybridized to Illumina HumanOmni1-Quad or Infinium Omni2.5–8 BeadChips. DNA samples were amplified, fragmented, and precipitated according to the manufacturer's protocol (Illumina). The precipitated DNA was hybridized BeadChips, incubated at 48 °C for 18 h. The BeadChips were washed and the hybridized oligos were subjected to single-base extension as per the Illumina protocol and scanned on the iScan (Illumina).

**Proteomic and phosphoproteomic sample preparation and data acquisition**

*Protein extraction and trypsinization*. PDXs profiled at the proteome level comprised 133 models, which included 2 technical replicates, a model with 2 different passages and 2 non-stable models. NSCLC PDX samples were weighted then sliced thinly. Tumors were lysed using a 0.02 mL lysis buffer per mg of wet tissue (0.5 M Tris pH 8.0, 50 mM NaCl, 2% SDS, 1% NP-40, 1% Triton X-100, 40 mM chloroacetamide, 10 mM TCEP, 5 mM EDTA) and then were sonicated for 15 s twice. The lysed samples were then heated at 95 °C for 20 min at 1000 rpm, then cooled to room temperature for 10 min. To remove cellular debris, samples were spun at $20,000 \times g$ for 5 min at 18 °C[95]. Protein concentration of the supernatant was measured by tryptophan fluorescence assay[96]. Proteins were extracted using methanol-chloroform protein precipitation and digested overnight at 37 °C with 1 µg trypsin/Lys-C mixture (Promega, Catalog No: V5073)[97].

*TMT labeling of peptides*. After overnight digestion, peptide concentration was measured using nanodrop at absorbance of 280 nm. As per manufacturer's instruction, 50 µg of each sample was labeled with 10-plex tandem mass tag reagents (ThermoFisher Scientific, Catalog: 90110). Eight PDX samples were grouped randomly and individually labeled with isobaric compounds (TMT10-127N, 127C, 128N, 128C, 129N, 129C, 130N, 130C), while two pooled mixtures of all PDX samples were labeled with TMT10-126 or 131, to serve as normalization controls[95]. Excess TMT label was quenched with 8% ammonium hydroxide prior to TMT-labeled samples being pooled at a 1:1:1:1:1:1:1:1:1:1 ratio. The pooled sample was then dried by vacuum centrifugation.

*Fractionation by HPLC*. The dried pooled sample was resuspended using 20 µL of ddH2O and was fractionated to 60 fractions using high pH reversed phase HPLC at 4 °C with a flow rate of 1 mL/min using Waters 1525 binary HPLC pump[98]. The 60 fractions were then dried with vacuum centrifugation and resuspended in 100 µL of 0.1% formic acid and transferred to a 96-well plate. Then each fraction was loaded into Evotip C18 trap column as per manufacturer's instruction. Samples were injected into Orbitrap Fusion Lumos MS using an Evosep One instrument using 30 samples per day setting.

*Total proteome MS acquisition settings*. For MS1 acquisition parameters, only ions with 2–6 charge states were accepted. MS1 acquisition resolution was 120,000 with automatic gain control (AGC) target value of $4 \times 10^5$ and maximum ion injection time (IT) of 50 ms for scan range of 550–1800 $m/z$. For MS2, ions were isolated in quadrupole by an isolation width of 2 $m/z$. MS2 scans were performed in the linear ion trap with maximum ion IT of 50 ms, AGC target value of $1 \times 10^4$, and normalized collisional energy (NCE) of 35 using the turbo scan rate. For MS3 scans, ions were isolated in the quadrupole with an isolation width of 2 $m/z$ and MultiNotch synchronous precursor selection MS3 scanning for TMT tags was used. Higher-energy collisional dissociation activation was used for fragmentation of ions with NCE of 65. Scans were then measured in the orbitrap using a 50,000 resolution for scan range of 100–500 $m/z$ with AGC target value of $1 \times 10^5$, and maximum ion IT of 50 ms. Dynamic exclusion of 65 s was used to permit more identification of peptides.

*Tandem phospoho-tyrosine (pY) peptide enrichment by covalent bound GST_Src_SH2 superbinder (sSH2)*. PDXs profiled at the pY level comprised 106 models which included 2 non-stable models. Total peptides were extracted as described in total proteomics step and quantified by Pierce peptide quantification kit (Pierce, Catalog No: 23275) according to manufacturer's instructions. One mg peptide of each sample was dried after trypsin digestion (Speed Vacuum) and dissolved in 200 µL of dissolving buffer from Thermo High-Select™ Fe-NTA Phosphopeptide Enrichment Kit. Phsopho-serine/threonine/tyrosine (pSTY) were enriched according to manufacturer's instructions. 95% of the final elution of pSTY peptides were dried and dissolved in 500 µL of Affinity Purification (AP) buffer (50 mM MOPS pH 7.2, 10 mM dibasic sodium phosphate, 50 mM NaCl). Fifty micrograms of Src-SH2 superbinder was used to enrich phosphor-tyrosine peptides according to our previous publication[99]. There were two improvements in the current experiment compared with previous publication. The first is, instead of using none-covalent bound His-Src-sSH2 fusion protein, GST-Src-sSH2 fusion proteins were cross-linked to glutathione beads with dimethyl pimelimidate (DMP). The second is using 60% of acetonitrile/0.1% trifluoroacetic acid[39], instead of 50 mM phenyl phosphate in AP buffer, to elute pY peptides from SH2 superbinder beads. The enriched pY peptides were then dried, dissolved in 100 µL of 0.1% formic acid and divided onto two Evotip C18 trap column for MS/MS analysis.

*MS analysis of pY peptide by CID and HCD-MS methods.* pY peptides enriched from PDX samples were analyzed by using an Orbitrap Fusion Lumos instrument. Samples were loaded by using EVOSEP tips and analyzed with 44 min MS runs as we have described previously[98]. Two separate LC-MS/MS runs were performed on every sample, the first one collected collision-induced dissociation (CID)-MS/MS spectra and the other one collected higher-energy collision dissociation (HCD)-MS/MS spectra. The parameters used for MS data acquisition of CID-MS/MS and HCD-MS/MS spectra were: (1) MS: top speed mode, cycle time = 3 s; scan range $(m/z)$ = 400–2000; resolution = 60,000; AGC target = 400,000; maximum injection time = 100 ms; MS1 precursor selection range = 700–2000; included charge states 2–6; dynamic exclusion after $n$ times, $n$ = 1; dynamic exclusion duration = 10 s; precursor priority = most intense; maximum intensity = 1E + 20; minimum intensity = 50,000; (2) CID-MS/MS: isolation mode = quadrupole; isolation window = 0.7; collision energy = 35%; detector type = Ion Trap; Ion Trap Scan Rate = Rapid, AGC target = 10,000; maximum injection time = 35 ms; Multistage Activation = True, Neutral loss mass = 97.9763; microscan = 1; (3) HCD-MS/MS: isolation mode = quadrupole; isolation window = 0.7; collision energy = 30%; stepped collision energy (%) = 5; detector type = orbitrap; resolution = 15,000; AGC target = 50,000; maximum injection time = 35 ms; microscan = 1.

**Reporting summary**. Further information on research design is available in the Nature Research Reporting Summary linked to this article.

## Data availability

PDX models are listed in the open global catalog of PDX models at the PDXFinder repository (pdxfinder.org). PDX models generated in this study are available for research use with institutional material transfer agreement. The mass spectrometry proteomics and phosphotyrosine data generated in this study have been deposited to the ProteomeXchange Consortium via the PRIDE partner repository[100] with the dataset identifier PXD016579 for total proteome and PXD016674 for phosphotyrosine. The gene expression data generated in this study has been deposited to GEO with the dataset identifier of GSE166999. DNA methylation data generated in this study has been deposited to EBI with the dataset identifier of E-MTAB-10156 proteome. The raw whole-exome sequencing data generated in this study are deposited to the European Genome-phenome Archive (https://ega-archive.org/) under Dataset ID: EGAD00001008601. The data are available under restricted access due to laws to protect the privacy of patients in alignment with University Health Network (UHN) Review Ethics Board (REB) approvals and individual patient informed consent forms. Access to the data can be obtained by qualified researchers as part of an academic or industry collaboration. Requests should include a research proposal indicating the intended use of data and planned analyses. Requests will be reviewed typically within two weeks by the UHN Data Access Committee (DAC) and should be made by using the DAC ID: EGAC00001000912. There are no time constraints on data access. Minimal essential PDX model characteristics have been deposited to the PDX Finder repository (https://www.pdxfinder.org/source/PMLB/). The publicly available DepMap sensitivity data28,29 used in this study are available through DepMap portal [https://depmap.org/portal/download/]. The publicly available data generated by Gillette et al.[12] is available via CPTAC data portal [https://cptac-data-portal.georgetown.edu/cptac/s/S056]. The publicly available data generated by Stewart et al.[14] was deposited to the ProteomeXchange Consortium via the PRIDE partner repository 79 with the dataset identifier PXD010429. The publicly available database dbSNP (version 138 flagged) can be found at https://ftp.ncbi.nlm.nih.gov/snp/, ExAC03 can be found at https://gnomad.broadinstitute.org/ and ESP6500 can be found at https://evs.gs.washington.edu/EVS/. The remaining data are available within the Article, Supplementary Information or Source Data file. Source data are provided with this paper.

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

## Acknowledgements

This work was supported by the Ontario Research Fund-Research Excellence grant RE-03–020 (MST), Canadian Cancer Society IMPACT grant #701595 (MST), Canadian Institutes of Health Research (CIHR) Foundation grant FDN-148395 (MST), and Project Grant (#364778, MFM), National Sciences and Engineering Research Council of Canada Discovery Grant (#RGPIN-2016-06305, MFM) and Princess Margaret Cancer Foundation. S. Mirhadi was funded by Restracomp. Drs. M. Cabanero and S. Sakashita were supported by the Terry Fox Foundation Special Training Initiative in Health Research at CIHR grant STP53912. M.-S. Tsao is the M. Qasim Choksi Chair in Lung Cancer Translational Research. G. Liu is the Alan B. Brown Chair in Molecular Genomics. F.A.S. is the Scott Taylor Chair in Lung Cancer Research. M.F. Moran is the Canada Research Chair in Molecular Signatures. We additionally thank Drs. Shaf Keshavjee, Gail Darling, Marcelo Cypel and Marc De Perrot for consenting their patients to contribute tumors for our xenograft project.

## Author contributions

Study conception and design, S.M., M.S.T., M.F.M.; Development of methodology, S.M., J.T., J.R.K., Q.L., N.A.P., N.M., K.I., K.H., B.J.G., S.S., M.C., S.N.M.F., M.L., P.T., M.S.T., and M.F.M.; Performed experiments and Acquisition of data, S.M., S.T., J.T., J.R.K., M.L., M.C., K.H., K.I., V.R., Q.L., S.N.M.F., and S.S.; Analysis and interpretation of data, S.M., S.T., N.A.P., N.M., Q.L., J.T., B.J.G., S.N.M.F., J.W., A.A.K., G.L., F.A.S. V.R.,Y.M., F.A.S., G.L., M.S.T. and M.F.M.; Writing of the manuscript, S.M., N.M., M.S.T. and M.F.M.; Provided Manuscript feedback, N.M., N.A.P., Q.L., S.A., B.J.G., S.N.M.F., J.W., F.A.S., G.L., G.Z., and T.K.W.; Administrative, technical, or material support: S.M., Q.L., M.L., N.A.P., K.Y., G.Z., and T.K.W.; Study supervision: M.S.T. and M.F.M.

## Competing interests

The authors declare no competing interests.
