## [Peer Review File · Nature Communications]

Integrative analysis of non-small cell lung cancer patient-derived xenografts identifies unique proteotypes associated with patient outcomesReviewers' Comments:

Reviewer #1:

Remarks to the Author:

Main findings:

This manuscript by Mirhadi and colleagues reports the multi-dimensional characterisation of patient-derived xenografts (PDXs) from non-small cell lung cancer (NSCLC). PDXs retained the transcriptomic and methylomic stratification developed in NSCLC patients for lung adenocarcinoma (LUAD) and lung squamous cell carcinoma (LUSC) and harboured genetic alterations analogous – in both nature and frequency – to those detected in patients. Proteomic profiling followed by unsupervised hierarchical clustering revealed three major clusters: one comprising mainly LUSC, a second comprising LUSC together with large cell neuroendocrine carcinoma, and a third comprising mainly LUAD. The authors also identified specific proteotypes that associated with defined transcriptomic or methylomic subgroups, had variable prognostic significance, and were enriched for selective actionable pathways and genomic alterations.

General assessment:

PDXs have been successfully deployed to extract multi-dimensional profiles with prognostic or predictive value, and also for biomarker/target validation or discovery. On this ground, the present study does a good job of gathering and integrating 'omics' data in a relatively large collection of models. Regrettably, the authors did not validate their findings in original (pre-implantation) tumors and did not attempt pharmacological targeting of the proposed vulnerabilities. As such, the paper remains descriptive and poorly informative.

Major concerns:

- 1) The transcriptomic, methylomic, genomic and proteomic data obtained in PDXs should have been benchmarked against analogous profiles in matched pre-implantation material. Is the assignment of individual PDXs to a specific transcriptional/epigenetic subgroup maintained in the corresponding original tumour? Are there differences in focal gene amplifications between fresh and mouse-passaged tumours when considering the number of gene copies? Is the overall genomic architecture preserved? An effort of this kind – which may be limited to a representative fraction of models – is necessary to strengthen the conclusiveness and generalisability of the reported observations.
- 2) It is quite surprising that the authors did not leverage the merit of PDXs – i.e., the possibility of in vivo therapeutic intervention based on molecular profiling – to target some of the druggable vulnerabilities emerged from their proteomic survey. Without this piece of information, it is difficult to anticipate whether the proposed liabilities are therapeutically relevant, hence potentially translatable to the clinical setting.

Reviewer #3:

Remarks to the Author:

NCOMMS-21-04521

Integrative analysis of non-small cell lung cancer patient-derived xenografts identifies unique proteotypes associated with patient outcomes

The authors test the ability to generate patient derived xenografts (PDXs) from 501 non-small cell lung cancer (NSCLC) specimens in NOD SCID mice, correlate these with clinical demographic features, and then perform proteogenomic analyses on the PDXs. They find a take rate of 27.3% and establish

137 NSCLC PDXs. The take rate was higher in lung squamous cancers (LUSCs, 64%) than lung adenocarcinomas (LUADs, 30%), and that patients whose tumors generated PDXs that could be passaged had inferior survival to those whose tumors could not be established as a propagating PDX. Their proteogenomic analyses for mRNA, proteomics, methylation and copy number changes, identified several different subgroups for LUAD and LUSC that were similar to prior published groups and to TCGA samples and several of these subgroups had survival differences. They also confirmed, in general, low correlations between the different profiling methods for quantitating the expression of individual genes. The proteome analyses also included phosphotyrosine evaluations. A variety of computational biology and pathway analyses were performed. In addition, they distinguished mouse from human proteins to identify stromal (mouse) components. Of note the PDXs were grown subcutaneously and not orthotopically. They conclude: "The models indicate 3 lung adenocarcinoma and 2 squamous cell carcinoma proteotypes that are associated with different patient outcomes, protein-phosphotyrosine profiles, candidate targets, and in adenocarcinoma, distinct stromal immune features. The PDX resource will foster proteome-directed stratification and development of new treatments for aggressive NSCLC. "No functional studies are presented.

Comments to Authors:

The work is technically well done and clearly presented. The large number of new PDXs derived by the authors with their proteogenomic characterization will be an important new lung cancer research resource if they are made freely available. All of the major findings the authors have made are confirmatory of many prior studies in lung cancer and other tumors. For example, the worse prognosis in patients whose tumors can form PDXs vs. those whose tumors do not generate PDXs. The various classification groups for LUAD and LUSCs from the PDXs and their relationship to tumor specimens are, essentially, confirmatory of several prior studies. The technology and methods for generating the PDXs, for obtaining the molecular and proteomic data, and computational analyses all are well known and standard and that is fine, but they do not "break new ground." While they derived prognostic signatures, these are essentially similar to prior reports. In addition, there is no independent validation population that was studied. So as a "resource" this has useful information and reagents for future studies.

1. However, in its present form, the manuscript as a "resource" provides data, but, in my opinion, the computational analyses should be structured in a way to provide a "roadmap" for future translational research by identifying the highest value targets, pathways, and PDXs to begin systematic testing for new lung cancer vulnerabilities. This latter information requires a more focused computational analysis rather than additional data.
2. As to additional data, there are three major questions the authors could have addressed but did not provide data on – all require some functional studies. I point these out, because providing answers to any of these three would have elevated this manuscript significantly. The first are the factors and mechanisms that allow some tumors to form PDXs while other, which can have devastating malignant behavior in patients do not. If we knew what any of these differences are, we would immediately have a much deeper understanding of lung cancer pathogenesis and potentially new therapeutic targets.
3. The second, what are the specific different vulnerabilities in any of the proteogenomic identified subgroups of NSCLC in these PDXs and do any of these provide a druggable therapeutic target and potential "therapeutic window"? While we may not have a final therapeutic is there a subset of these PDXs that could be "cured" in preclinical studies by targeting something that their proteogenomic studies identified, using, for example, functional genomics (e.g., drop out screens) and are there precision medicine biomarkers to identify ahead of time which PDXs would respond to this specific targeting and which would not? The DepMap studies are all done in vitro and all need to be verified in vivo (e.g., in xenografts). One could imagine that their proteogenomic studies identified a subgroup of LUADs for which we don't have targeted therapy, and the specific gene/protein to target within this group. For example, there could be a phosphotyrosine target that is a dependency.

4. Finally, the tumors were studied subcutaneously in NOD-SCID mice and there are probably some differences in the tumor microenvironment (TME). Information on whether the TME was the same orthotopically (in the lung) and in subcutaneous tissues would be important. However, ultimately, we would want to know that a specific PDX did something to the TME that enabled "immune escape" of the tumor, and/or now became a potential therapeutic target to aid in immunotherapy. For example, the tumors in this subgroup made some specific cytokine that could be targeted (I just give this as an example).

5. Thus, the PDXs and the proteogenomic analyses would give us not just a "resource" but an example of the importance of this resource. I would leave it to the authors to decide which if any or all of these examples they could provide data on to make this publication go beyond reporting of a new panel of PDXs.

Reviewer #4:

Remarks to the Author:

In this manuscript, Mirhadi et al presented a large-scale proteogenomic study of non-small cell lung cancer PDX samples. They conducted the genomic, transcriptomic, proteomic and tyrosine phosphoproteomic analysis of the PDX model samples that were generated from NSCLC patients. The authors showed that transcriptome-based subtypes of the PDX samples were similar and comparable to the previously reported subtypes of patients in TCGA cohorts, although the frequencies of some well-known DNA alterations in NSCLC were different between PDX samples and patients. Proteomic analysis showed distinct LUAD and LUSC proteotypes, which were associated with different prognosis and molecular features. The authors further nominated potential proteotype-specific druggable targets and biomarkers. Finally, the authors further investigated some mouse (stromal) proteome features among the proteotypes.

Overall, the authors conducted a huge amount of work in NSCLC PDX models, which could potentially serve as a valuable resource for NSCLC research. However, this manuscript does not provide enough details on their mass spectrometry data processing procedure and the following proteomics analysis. Therefore, it is hard to evaluate the reliability of their conclusions drawn from the proteomics data. In addition, although the authors generated a lot of data, many of their analyses and results are descriptive and preliminary.

Specific comments:

1. In this manuscript, different number of samples were used for different analysis. It is very confusing how many samples were used in each analysis. For example, this study generated 137 PDX models. Why the authors only conducted proteomic analysis of 133 samples (Only 133 proteome data shown in supplementary table and supplementary figure 2A)? The authors performed phosphoproteomic analysis of 125 PDX samples, but they only showed 115 phosphoproteome data in the manuscript. This study generated 65 LUSC PDX models and 58 LUAD PDX models. Only 60 LUSC PDX samples and 58 LUAD PDX samples were used for unsupervised consensus clustering and PCA analysis. The number of proteins, phosphoproteins, phosphosites identified in each sample should be provided.

2. Since the engraftment rate for stable PDX is low (27.3%), the representation of PDX samples are obviously quite different (distorted) from the population of NSCLC patients. Although the authors compared their genomics data with those of primary tumors in TCGA cohorts, the manuscript did not present a clear general view on the genetic and transcriptomic differences (over- or under-representation) between the PDX samples and clinical patients. This is an important clue for the proper use of PDX models in NSCLC study.

3. Quality control processes for mass spectrometry data processing and details for bioinformatic analysis should be provided (e.g. quantification reliability, missing values, proteins used for clustering, etc.). The samples are mixtures of human and mouse proteins. The authors should clearly clarify this

issue and explain how the data was used for subsequent analysis. Without these details, the reliability of the proteomic results is difficult to be evaluated.

4. Why did the authors use different quantification approach for proteome (TMT labeling) and tyrosine phosphoproteome (label free). Without strict quality control, affinity enrichment-based label free quantification is not reliable. In addition, the data quality of phosphotyrosine proteome seems to be relatively low. For most of the samples, only less than 100 pY sites were quantified. How did the authors use this data with such a low number of phosphosites for further analysis?

5. The genetic backgrounds between NSCLC patients from the European ancestry (Nature. 2014, 543-550; Nat Genetics 2016, 607) and East-Asian ancestry (Nat Genet. 2020, 52:177; Cell. 2020 182:226; Cell. 2020,182:245) are quite different. For example, their transcriptome subtypes are different (Nature. 2014, 543-550; Nat Genet. 2020, 52:177). The authors should provide the ethnic information of the human subjects and compared their data with the proper patients.

6. For tyrosine phosphoproteome, did the authors identify the substrates in RTK/RAF/RAS pathways, which are frequently altered in NSCLC? The authors need clearly present this information for each sample in their manuscript.

7. A compelling advantage of PDX models is that they can be used for drug efficacy test. The authors nominated some proteotype-specific drug target based on the enrichment pathways from proteome and pY proteome data, but without any experimental validation. I do not think such analysis could provide convincing information.

8. For potential biomarker identification, what is the rationale for 4-fold difference in expression? What is the frequency of each protein occurred in the samples? The authors should provide more information for the potential biomarkers. Validation in clinical samples is also necessary. What is the potential application of these biomarkers?

9. The PDX models were from severely compromised immunodeficient mice, which were significantly different from the human cancer microenvironment. The authors analyzed some mouse proteome features (defined them as stromal) among the proteotypes in their manuscript. However, this mouse proteome is fundamentally different from real stroma in human. I do not think such analysis could reflect the real human tumor microenvironment.

Reviewer comments are reproduced verbatim in bold, and new text that has been added to the manuscript is highlighted in italics.

Reviewer #1

1) The transcriptomic, methylomic, genomic and proteomic data obtained in PDXs should have been benchmarked against analogous profiles in matched pre-implantation material. Is the assignment of individual PDXs to a specific transcriptional/epigenetic subgroup maintained in the corresponding original tumour? Are there differences in focal gene amplifications between fresh and mouse-passaged tumours when considering the number of gene copies? Is the overall genomic architecture preserved? An effort of this kind – which may be limited to a representative fraction of models – is necessary to strengthen the conclusiveness and generalisability of the reported observations.

We agree with the reviewer on the importance of comparing the omics profiles of matched pre-implantation tumors and the PDXs. We have previously published such comparison in subsets of the models (PMID: 25429762, 27750381) and regret that this information was not clearly conveyed in the original manuscript. In the revised manuscript we have modified the introduction to provide this missing information. These modifications are described below in the excerpt from the revised manuscript:

Introduction, page 3:

The engrafted patient tumors retain the phenotypic features of the primary tumors, including histology, mutational landscape, RNA and protein expression 20–25. We previously measured high correlations ($r_s > 0.67$) between 11 PDX and their matched patient NSCLC for individual profiles of their DNA copy number, mRNA and protein abundances²⁶. In another study comparing 36 matched PDX-primary NSCLC we demonstrated retention of $>90\%$ of SNP mutations and the close recapitulation in PDX models of gene expression, methylation, and protein-phosphotyrosine (pY) profiles²⁷.

2) It is quite surprising that the authors did not leverage the merit of PDXs – i.e., the possibility of in vivo therapeutic intervention based on molecular profiling – to target some of the druggable vulnerabilities emerged from their proteomic survey. Without this piece of information, it is difficult to anticipate whether the proposed liabilities are therapeutically relevant, hence potentially translatable to the clinical setting.

We agree with the reviewer on the importance and potential to undertake pre-clinical investigations of newly predicted, proteotype-associated therapeutic vulnerabilities by using the PDX models. While the systematic testing of identified candidate targets is beyond the scope of this publication, we have revised the manuscript to address this concern and further demonstrate the potential for proteomics to complement genomics-based hypotheses of therapeutic vulnerabilities. The revised manuscript includes new data that demonstrate the pre-clinical utility of the PDX models related to receptor tyrosine kinases (RTKs) and tyrosine kinase inhibitors (TKIs). The revised text is shown below:

Results, page 17:

EGFR tyrosine kinase inhibitors (TKIs) are currently restricted to cases with EGFR hotspot mutations. We observed that cases with hotspot mutations do not always have high EGFR expression nor increased activated pY sites compared to cases without mutations (Fig. 6C). Herein, we demonstrate that a PDX with WT amplification and high EGFR protein expression and pY enrichment but no oncogenic hot spot mutation significantly responded to treatment with the EGFR TKI Afatinib (p -value <0.0001) (Fig. 7E). This is an example wherein proteome analysis might reveal potentially responsive, activated target pathways not uncovered by genome/transcriptome analyses. In a previous study, we treated 4 PDX models bearing EGFR activating mutations by multiple EGFR inhibitors⁵⁵. Two models (PHLC137 and 192) (Fig. 7C-indicated by blue diamond) responded to Erlotinib, Dacomitinib, Afatinib and Cetuximab, whereas the other two models either did not, or only responded to cetuximab (PHLC148 and 164) (Fig. 7C-indicated by yellow star). Comparing the proteome of the two responders vs. non-responders revealed VIM expression significantly higher in the non-responders, consistent with previous reports⁵⁶, and CALML3 expression higher in responders

(Supplementary Fig. 7A). Another potential TKI target in NSCLC is fibroblast growth factor receptor 1 (FGFR1), which is amplified in 20% of LUSC and known to be involved in cell proliferation and survival. Indeed, 24% (14 of 58) of our LUSC cases contain an amplification of 8p11.23 that includes FGFR1 (Supplementary Fig. 7B). However, we saw no significant change relative to non-amplified samples in the levels of FGFR1 protein or protein-pY in these cases (Supplementary Fig. 7C). Consistent with these observations, we tested the FGFR1 inhibitor BGJ398 for growth inhibition in four randomly-selected FGFR1-amplified tumors (PHLC-200,-274, -299, and -321), and found that in all cases there was an initial minor shrinkage of tumors but ultimately no inhibition of PDX tumor growth (Supplementary Fig. 7D-G). These examples demonstrate the utility of the PDX models to test therapeutic hypotheses including target validation based on protein expression or pathway activation.”

Reviewer #3

1. In its present form, the manuscript as a “resource” provides data, but, in my opinion, the computational analyses should be structured in a way to provide a “roadmap” for future translational research by identifying the highest value targets, pathways, and PDXs to begin systematic testing for new lung cancer vulnerabilities. This latter information requires a more focused computational analysis rather than additional data.

We appreciate the reviewer’s suggestion to include a more focused computational analysis and a roadmap for future translational work. To address this comment, we have added to the manuscript a more focused analysis to identify the highest value targets and pathways as detailed below in the excerpt from the revised manuscript. In addition, we have revised Figure 1 to include a schematic roadmap that presents the steps taken to identify, validate, and prioritize targets and pathways as a guide for future translational research in lung cancer.

Legend to Fig. 1:

Figure 1. A roadmap to cancer proteotype discovery and utility. A subset of 137 of 501 primary NSCLC tumors engrafted to yield PDX models. PDXs represent the most aggressive subset of NSCLC and were profiled for gene expression, gene copy number variation, DNA methylation, exome mutations, proteome and phosphotyrosine(pY)-proteome. Proteome profiling revealed proteotypes associated with patient survival differences. Proteotypes display distinctive active pathway features and associated candidate therapeutic targets. Signatures comprising proteotype markers effectively stratify orthogonal NSCLC primary tumors 13,15 as well as NSCLC DepMap cell lines 26, which enables a degree of candidate target validation and prioritization based on alignment with DepMap sensitivities 27.

Added to the Results, page 15:

The proteotype protein signatures were also able to effectively categorize 34/34 LUAD and 9/12 LUSC cell lines previously characterized at the proteome level as part of the DepMap project²⁶. The LUAD cell lines clustered into groups corresponding to LUAD1 (8 lines), LUAD2 (13 lines), and LUAD3 (13) (Supplementary Fig. 6A-B), and the LUSC cell lines clustered into groups corresponding to LUSC1 (6 lines) and LUSC2 (3 lines) (Supplementary Fig. 6C-D). Genetic and pharmacological sensitivities associated with the cell lines grouped by proteotype were defined by using DepMap data²⁷ (Supplementary Table 11). This analysis revealed proteotype-specific sensitivities (Supplementary Table 11), including top set of candidate actionable targets and molecules based on the effect size of their inhibition on cell line viability (Supplementary Fig. 6E-F). We further identified sensitivities that matched significantly differential proteins of the proteotypes (Supplementary Table 11). Pathway analysis of these matched targets showed the LUAD1 lines to be sensitive to losing components of the TCA cycle, LUSC1 spliceosome and LUSC2 ribosome biogenesis. This was consistent with the enrichment of high activity of TCA cycle in LUAD1 and higher enrichment and activity of spliceosome in LUSC1 (Supplementary Table 11). Ribosome components were expressed lower in LUSC2, which might be why LUSC2 cell lines are sensitive to losing components of ribosome biogenesis.

2. As to additional data, there are three major questions the authors could have addressed but did not provide data on – all require some functional studies. I point these out, because providing answers to any of these three would have elevated this manuscript significantly. The first are the factors and mechanisms that allow some tumors to form PDXs while other, which can have devastating malignant behavior in patients do

not. If we knew what any of these differences are, we would immediately have a much deeper understanding of lung cancer pathogenesis and potentially new therapeutic targets.

We thank the reviewer for raising these important questions, which we have also been considering, especially on the factors and mechanisms associated with engraftment that may be related to tumor aggressiveness and patient outcomes. Addressing this question will require a multi-omics analysis of engrafting and non-engrafting primary patient tumors. This is the subject of our ongoing research and represents a major line of research that is beyond the scope of this large study in which we have generated and comprehensively analyzed 137 patient-derived xenograft tumors.

3. The second, what are the specific different vulnerabilities in any of the proteogenomic identified subgroups of NSCLC in these PDXs and do any of these provide a druggable therapeutic target and potential “therapeutic window”? While we may not have a final therapeutic is there a subset of these PDXs that could be “cured” in preclinical studies by targeting something that their proteogenomic studies identified, using, for example, functional genomics (e.g., drop out screens) and are there precision medicine biomarkers to identify ahead of time which PDXs would respond to this specific targeting and which would not? The DepMap studies are all done in vitro and all need to be verified in vivo (e.g., in xenografts). One could image that their proteogenomic studies identified a subgroup of LUADs for which we don’t have targeted therapy, and the specific gene/protein to target within this group. For example, there could be a phosphotyrosine target that is a dependency.

This comment is akin to comment 2 of reviewer 1. We fully agree with the reviewer of the important potential utility of the PDX models to test emerging therapeutic hypotheses. Systematic screening of our PDX models for therapeutic vulnerability is a huge ambitious project, which will likely require extensive and collaborative efforts among lung cancer researchers. We hope this effort will be stimulated by publication of our findings as a timely resource. We have added a statement to the Discussion to describe the technical limitation of the PDX system testing for RNAi or CRISPR screens:

“Another obvious limitation of the PDX models is that they are not readily feasible for comprehensive/systems type analyses of gene dependencies or chemical screens.”

To address the reviewer’s concerns, we have added new data, which demonstrate the pre-clinical utility of the PDX models related to receptor tyrosine kinases (RTKs) and tyrosine kinase inhibitors (TKIs). We also included new analysis of the DepMap screens to address the proteotype-specific sensitivity comment. The follow text has been added to the revised manuscript:

Results, page 17:

EGFR tyrosine kinase inhibitors (TKIs) are currently restricted to cases with EGFR hotspot mutations. We observed that cases with hotspot mutations do not always have high EGFR expression nor increased activated pY sites compared to cases without mutations (Fig. 6C). Herein, we demonstrate that a PDX with WT amplification and high EGFR protein expression and pY enrichment but no oncogenic hot spot mutation significantly responded to treatment with the EGFR TKI Afatinib (p-value<0.0001) (Fig. 7E). This is an example wherein proteome analysis might reveal potentially responsive, activated target pathways not uncovered by genome/transcriptome analyses. In a previous study, we treated 4 PDX models bearing EGFR activating mutations by multiple EGFR inhibitors⁵⁵. Two models (PHLC137 and 192) (Fig. 7C-indicated by blue diamond) responded to Erlotinib, Dacomitinib, Afatinib and Cetuximab, whereas the other two models either did not, or only responded to cetuximab (PHLC148 and 164) (Fig. 7C-indicated by yellow star). Comparing the proteome of the two responders vs. non-responders revealed VIM expression significantly higher in the non-responders, consistent with previous reports⁵⁶, and CALML3 expression higher in responders (Supplementary Fig. 7A). Another potential TKI target in NSCLC is fibroblast growth factor receptor 1 (FGFR1), which is amplified in 20% of LUSC and known to be involved in cell proliferation and survival. Indeed, 24% (14 of 58) of our LUSC cases contain an amplification of 8p11.23 that includes FGFR1 (Supplementary Fig. 7B). However, we saw no significant change relative to non-amplified samples in the levels of FGFR1 protein or protein-pY in these cases (Supplementary Fig. 7C). Consistent with these observations, we tested the FGFR1 inhibitor BGJ398 for growth

inhibition in four randomly-selected FGFR1-amplified tumors (PHLC-200,-274, -299, and -321), and found that in all cases there was an initial minor shrinkage of tumors but ultimately no inhibition of PDX tumor growth (Supplementary Fig. 7D-G). These examples demonstrate the utility of the PDX models to test therapeutic hypotheses including target validation based on protein expression or pathway activation.

...and Results, page 15:

The proteotype protein signatures were also able to effectively categorize 34/34 LUAD and 9/12 LUSC cell lines previously characterized at the proteome level as part of DepMap project²⁶. The LUAD cell lines clustered into groups corresponding to LUAD1 (8 lines), LUAD2 (13 lines), and LUAD3 (13) (Supplementary Fig. 6A-B), and the LUSC cell lines clustered into groups corresponding to LUSC1 (6 lines) and LUSC2 (3 lines) (Supplementary Fig. 6C-D). Genetic and pharmacological sensitivities associated with the cell lines grouped by proteotype were defined by using DepMap data²⁷ (Supplementary Table 11). This analysis revealed proteotype-specific sensitivities (Supplementary Table 11), including a top set of candidate actionable targets and molecules based on the effect size of their inhibition on cell line viability (Supplementary Fig. 6E-F). We further identified sensitivities that matched significantly differential proteins of the proteotypes (Supplementary Table 11). Pathway analysis of these matched targets showed the LUAD1 lines to be sensitive to losing components of the TCA cycle, LUSC1 spliceosome and LUSC2 ribosome biogenesis. This was consistent with the enrichment of high activity of TCA cycle in LUAD1 and higher enrichment and activity of spliceosome in LUSC1 (Supplementary Table 11). Ribosome components were expressed lower in LUSC2, which might be why LUSC2 cell lines are sensitive to losing components of ribosome biogenesis.

4. Finally, the tumors were studied subcutaneously in NOD-SCID mice and there are probably some differences in the tumor microenvironment (TME). Information on whether the TME was the same orthotopically (in the lung) and in subcutaneous tissues would be important. However, ultimately, we would want to know that a specific PDX did something to the TME that enabled “immune escape” of the tumor, and/or now became a potential therapeutic target to aid in immunotherapy. For example, the tumors in this subgroup made some specific cytokine that could be targeted (I just give this as an example).

We agree and appreciate the importance of tumor microenvironment and its recapitulation in the PDX system. We agree that the TME of PDX tumors grown in NOD-SCID mice will differ from that of primary tumors and that PDXs grown subcutaneously may differ from those grown orthotopically. We focused on our established subcutaneous PDX protocol with the rationale that in this system there is strong and significant prognostic impact associated with engraftment, which has not been demonstrated in an orthotopic model. We are in the process of determining the feasibility of orthotopic PDX models of primary NSCLC, but this protocol is not yet established and remains beyond the scope of the current study. Although it is possible to identify the presence of certain immune cells in primary tumors based on expression of cell-type specific markers, fibroblast stromal signatures are hard to identify and assignment of molecular features as stroma- or tumor-derived is not readily achieved by analysis of bulk tumor tissue. For these reasons, comparison of human vs mouse stroma is a challenging task and never been done before successfully. In term of immune-deficient host, we are also working on a strategy to enable use of PDX for immune-oncology research, including the use of a humanized mouse model or other approaches. However, for this manuscript, we have added in our discussion the shortcoming of using our model:

Discussion, page 22:

An obvious drawback of our method was the use of subcutaneous implantation of patient tumors in NOD-SCID mice that have a compromised immune system. NOD-SCID mice have a reduced innate immunity and nearly no adaptive immunity^{74,75}. Although, these immunodeficient features are necessary to prevent tumor rejection, this limits the scope of stromal differences that would usually be present in an immune-healthy individual. This also makes NOD-SCIDs not the ideal model for testing of immuno-modulatory treatments.

While these drawbacks exist, the intact part of the immune system of these models and the murine origin of the stromal component, allows us to clearly see differences among two of our proteotypes, LUAD1 and LUAD3, which highlight the influence of tumor signaling can have on modulating immune components of the tumor cells or of the mouse.

To address the second part of Reviewer #3's comment regarding potential immune-modulatory proteins that could be targeted, we note that we identified several highly expressed immunomodulatory proteins in LUAD3 known to be important for immune suppression, see results page 19. LGALS1, expressed 11.7 times higher in LUAD3 is tumor derived and secreted into the ECM where it induces apoptosis in anti-tumor immunocytes and skews the cytokine milieu towards promoting tumor growth¹⁶. Another key immunosuppressive protein expressed 3.5 higher in LUAD3 is NT5E, which is a membrane bound protein that converts extracellular AMP to adenosine, a metabolite which in turn restricts inflammatory immune response through negative feedback loop on adenosine receptor expressing neutrophils¹⁷. Higher expression of such immunomodulatory proteins in this proteotype might explain the differential recruitment of TME.

Reviewer #4

1. In this manuscript, different number of samples were used for different analysis. It is very confusing how many samples were used in each analysis. For example, this study generated 137 PDX models. Why the authors only conducted proteomic analysis of 133 samples (Only 133 proteome data shown in supplementary table and supplementary figure 2A)? The authors performed phosphoproteomic analysis of 125 PDX samples, but they only showed 115 phosphoproteome data in the manuscript. This study generated 65 LUSC PDX models and 58 LUAD PDX models. Only 60 LUSC PDX samples and 58 LUAD PDX samples were used for unsupervised consensus clustering and PCA analysis. The number of proteins, phosphoproteins, phosphosites identified in each sample should be provided.

We regret the confusion surrounding numbers of samples and analyses that was apparent in the original submission. In brief, our study generated 137 stable PDX models but the number of PDX models profiled by each platform varied due to availability of tumor tissues. To clarify and avoid confusion, we have modified the main text in the Methods section to include as a first statement the number of PDX samples that were profiled by that platform. We have also modified Table S8-Clinical&OmicSubtypeInfo to include all 6 platforms and provided information on which platforms were used to profile per each model. For each subsequent analysis, all relevant models profiled by that platform have been used.

To address the second comment, we have included 3 additional columns to Table S8-ProteomeExperimentalInfo tab to include information on number of quantified proteins, phosphotyrosine peptides and phosphoproteins per model.

2. Since the engraftment rate for stable PDX is low (27.3%), the representation of PDX samples are obviously quite different (distorted) from the population of NSCLC patients. Although the authors compared their genomics data with those of primary tumors in TCGA cohorts, the manuscript did not present a clear general view on the genetic and transcriptomic differences (over- or under- representation) between the PDX samples and clinical patients. This is an important clue for the proper use of PDX models in NSCLC study.

We appreciate the reviewer's concerns regarding comparisons of genetic and transcriptomic features between the PDX models and patient primary tumors. We have established that the PDX models represent NSCLC tumors with more aggressive behavior and carry poorer prognosis for the patient, and therefore may represent advanced stage patient tumors. However, currently there is little or no comprehensive omics data on the latter that we can compare with our PDX omics data. As mentioned in our response to comment 1 of Reviewer #1, we have first prioritized the profiling of the PDX models as we consider their characterization as essential in order to validate them and enable/foster further studies based on them. However, our previous publications have partially addressed the PDX/clinical tumor comparison (Please see response to Reviewer #1, comment 1). Nevertheless, we note that in this manuscript we showed that the landscape of transcriptomic differences reported between patient tumors (TCGA) are fully represented in our cohort of PDXs and in the same proportions indicating that the PDX models may have utility as models for specific subtypes of omics features found in primary NSCLC.

We have added additional information to the revised manuscript discussing this issue for the methylation subtypes and rephrased to make clearer the differences in genetic alterations as follow:

Results, page 7:

The data suggests that DNA methylation signatures are also conserved and represented in the PDX models. Although the frequency of methylation-based subtypes was on par with that of primary LUAD, in LUSC the C1 subtype was under-represented by 50% and the C2 subtype was over-represented by 100% in comparison to a patient population¹⁴. These suggest that biological aspects related to these subtypes might influence engraftment.

...and:

We observed that alterations frequently identified in LUAD and LUSC primary tumors are represented in the PDX models but in some cases were over- or under-represented compared to the frequencies seen in patient tumor populations³¹ (Fig. 3A-B) (Supplementary Table 8). For instance, KRAS mutations are over-represented in LUAD PDXs at 50% frequency compared to 30% in patient primary tumors, whereas EGFR sensitizing mutations in exon 18-21 are under-represented³¹ (Fig. 3A, Supplementary Table 8). These findings are consistent with mutant KRAS being associated with better and mutant EGFR poorer engraftments, respectively, hence poorer and better prognoses^{10,30,32}. NFE2L2, FAT1 and NOTCH1, were over-represented in LUSC PDXs compared to primary tumors³¹, suggesting such alterations might favor more aggressive cancer phenotype and PDX formation. CNV in LUAD and LUSC PDX tumors mirrors closely primary tumors, (Fig. 3C, D) (Supplementary Table 8). Overall, these analyses revealed that the NSCLC PDX models retain genomic features that resemble primary tumors including some that may be related to aggressiveness and engraftment.

3. Quality control processes for mass spectrometry data processing and details for bioinformatic analysis should be provided (e.g. quantification reliability, missing values, proteins used for clustering, etc.). The samples are mixtures of human and mouse proteins. The authors should clearly clarify this issue and explain how the data was used for subsequent analysis. Without these details, the reliability of the proteomic results is difficult to be evaluated.

We appreciate the reviewers request for these important technical details that were not readily accessible in the original submission. This information is now included in the revised manuscript as follows.

Results, page 8:

Tandem mass tag (TMT)-based quantitative MS analysis of PDX tumors was undertaken (Supplementary Fig. 2A). For data quality assurance, a replicate sample pair in the same experimental group and two pairs of replicates samples split into different experimental groups were analyzed. The technical replicates provided a readout for fidelity of the normalization method and technical robustness. A strong linear relationship between the replicates was seen for each pair ($R^2 \geq 0.94$) (Supplementary Fig. 2E). PCA verified that samples did not cluster based on experimental group or isobaric labels (Supplementary Fig. 2F-G). MS analysis of 133 PDX samples uncovered a total of 13284 proteins using a strict false discovery rate (FDR) of 0.01 of which 6830 were identified as human, 4423 as mouse, and 2031 that did not contain unique human or mouse peptides and therefore were assigned as human/mouse (Fig. 4A). To assess tumor-stroma composition, the fraction of total ion intensity corresponding to human, mouse, and human/mouse proteins was determined for each PDX sample (Fig. 4B) (Supplementary Table 8). This provided a unique opportunity to correct for discrepancies in tumor (i.e. human) cell composition across samples, which ranged between 20-70% (Fig. 4B) (Supplementary Table 8). This ensured that measured changes in protein abundance reflects proteome remodeling in tumor cells and not differences in tumor cellularity.

4. Why did the authors use different quantification approach for proteome (TMT labeling) and tyrosine phosphoproteome (label free). Without strict quality control, affinity enrichment-based label free quantification is not reliable. In addition, the data quality of phosphotyrosine proteome seems to be relatively low. For most of the samples, only less than 100 pY sites were quantified. How did the authors use this data with such a low number of phosphosites for further analysis?

We agree with the reviewer that it would have been ideal if feasible to use identical technical platforms for whole proteome and pY proteome analyses. However, since we were specifically interested in capturing pY sites, and these are very low in abundance, we needed to start from the highest amount of tumor tissue, which based on tissue availability was typically 1 mg. TMT-labeling 1mg of peptide was beyond our budget and labeling minute amounts of affinity purified phosphopeptides was beyond our technical capability. In addition, we had concerns of reports of significant neutral loss and reduced proton mobility of TMT-labelled pY-peptides has been reported, which might have distorted our findings (Everley et al....and Gygi SP. Neutral Loss Is a Very Common Occurrence in Phosphotyrosine-Containing Peptides Labeled with Isobaric Tags. *J Proteome Res.* 2017, 16(2):1069-1076, PMID: 27978624). Therefore, we strived to optimize rather than integrate our respective pY and whole proteome workflows.

We note that our yields of pY peptides from 1 mg (as protein) of unstimulated (for example by growth factor treatment or pY phosphatase inhibition with sodium orthovanadate) starting material is comparable with publications that used similar strategies. Previous efforts by other groups quantified >100 pY sites from 3 mg of starting material 18. Another study identified 1800 pY sites from 2 mg of stimulated treated Jurkat T cells but only 343 pY sites from 5 mg of unstimulated cells. Finally, most comparable to our starting material, another study identified 197 pY sites from 5 mg of heterogeneous tissue sample 19. Another study done on heterogeneous tissue identified 845 pY sites from 10 mg of protein 20. Altogether, these show that our method and quantification was comparable to other studies, and we only lacked in starting material, which is always a limitation when working with tumor tissue.

Study	Sample	Protein amount	# of pY
This PDX study	Heterogeneous PDX tumors	1 mg	~130/sample
Yao et. al., 2019 ¹⁸	Cell line-unstimulated	1-3 mg	~100/sample
Dong et. al., 2017 ¹⁹	Cell line-stimulated	2 mg	1800
	Cell line-unstimulated	5 mg	343
	Heterogeneous tissue	5 mg	197
Jedrychowski et. al., 2011 ²⁰	Heterogeneous tissue	10 mg	845

Prior to conducting this study, to ensure that our sample-preparation and pY capturing strategy is reliable and comparable to similar studies. We benchmarked our method and compared it to other studies. Using the two-step enrichment strategy used in this study we were able to pull down ~1400 pY sites from 1mg of EGF or vanadate-treated HeLa cells. Since the number of pY sites identified in the PDX models were limited, we did not intend to use this data for subtyping samples. Instead, once the proteotypes were identified by using the total proteome analysis, pY phosphosites were compared across proteotype groups to identify differential pY sites.

5. The genetic backgrounds between NSCLC patients from the European ancestry (Nature. 2014, 543-550; Nat Genetics 2016, 607) and East-Asian ancestry (Nat Genet. 2020, 52:177; Cell. 2020 182:226; Cell. 2020,182:245) are quite different. For example, their transcriptome subtypes are different (Nature. 2014, 543-550; Nat Genet. 2020, 52:177). The authors should provide the ethnic information of the human subjects and compared their data with the proper patients.

We agree with the reviewer that ethnic background can provide us with an additional dimension to view whether differences seen are a consequence of difference in ethnicity background. However, at our institution (University Health Network, Toronto, Canada) and generally in Canada, patient ethnicity data is not systematically collected in the medical record, as there was no standard or reliable approach to record patient ancestry.

6. For tyrosine phosphoproteome, did the authors identify the substrates in RTK/RAF/RAS pathways, which are frequently altered in NSCLC? The authors need clearly present this information for each sample in their manuscript.

We agree with the reviewer that since the tumor-associated RTK/RAS/RAF phosphoproteome is frequently altered in NSCLC this information is relevant and would improve our study. To address this concern, we have included a new excel sheet in Supplementary Table 10 (pY functional category) where all phosphosites are grouped into their functional category of receptor tyrosine kinases (RTK), phospho-tyrosine kinases (PK-Y), phosphokinases (PK), proteins with phosphotyrosine binding domains (SH2/PTB), signaling proteins, and others. Here, the relative signal for each case and per proteotype has been color coded and additionally the raw signals can also be viewed. We have also provided information per peptide for pathway involvement based on KEGG and Reactome, cellular component based on GO annotation (GOCC), and protein family based on Panther family (pfam), and whether these might be part of a protein complex based on Corum protein complex.

7. A compelling advantage of PDX models is that they can be used for drug efficacy test. The authors nominated some proteotype-specific drug target based on the enrichment pathways from proteome and pY proteome data, but without any experimental validation. I do not think such analysis could provide convincing information.

This comment is akin to comment 2 of reviewer #1. We agree on the importance of testing therapeutic predictions in the PDX models, which will require extensive and likely collaborative efforts among lung cancer researchers. This is a major motivation for our timely study. To address the reviewer's concern, we have added new data, which demonstrate the pre-clinical utility of the PDX models related to receptor tyrosine kinases (RTKs) and tyrosine kinase inhibitors (TKIs). We also modified our analysis to use the DepMap screens to identify the highest value sensitivities including sensitivities that matched differentially expressed proteins of our proteotypes. These new data are described in the revised manuscript as indicated above in our response to question 3 from Reviewer #3.

8. For potential biomarker identification, what is the rationale for 4-fold difference in expression? What is the frequency of each protein occurred in the samples? The authors should provide more information for the potential biomarkers. Validation in clinical samples is also necessary. What is the potential application of these biomarkers?

We thank the reviewer for this comment and regret that this information was not clearly conveyed in the original manuscript. The goal of selecting markers for the proteotypes was to identify a smaller set of proteins that could be used to survey external cohorts in order to stratify them according to our defined proteotypes. A smaller number of markers was sought in order to be compatible with targeted approaches for detection such as immunohistochemistry (IHC) or parallel reaction monitoring MS. To identify a smaller subset, we used a 4-fold cut-off among the identified significantly altered proteins, a grouping that considered variance, frequency, and fold change. A fold-difference cut-off has been shown to be a reliable and reproducible method with more resistance to outliers 21. Previous efforts have demonstrated that with a fold-change difference >4 in TMT-acquired proteome data the correlation of MS signal to western blot intensity is very high, with a Pearson r between 0.8-1.21. This is perhaps because the higher the fold difference the higher the likelihood that it will surpass technical and technological limitations that introduce noise in different surveying methods i.e., percent stromal contamination, method/machine variation/differences, and detection sensitivity. In this manuscript, we used proteotype markers defined by the 4-fold parameter in order to successfully survey two independent clinical NSCLC cohorts of ~ 100 patients each for 'clinical validation' (Supplementary Fig. 5). These markers successfully grouped the external patient cohorts in what we defined as our proteotypes, which matched the bona fide subtypes identified in the original studies, demonstrating the clinical utility of these signature protein markers. We further used these markers to assign proteotypes to DepMap LUAD and LUSC cell lines as a rationale to identify proteotype sensitivities to drugs and knockdown/knockouts in the DepMap dataset (Supplementary Fig. 5).

To address the reviewers' concerns, we have included an additional tab to Supplementary Table 9-proteotype biomarkers that includes all proteotype markers along with information on frequency, significance, fold change, and localization. We have further modified the text of the manuscript to describe our rationale for the strategy:

Results, page 13:

In order to identify protein signatures that could be used to define the proteotype of primary tumors, we considered only proteins significantly differentially expressed (≥ 4 -fold) in proteotypes and detected in at least 50% of cases (Supplementary Fig. 4) (Supplementary Table 9). This threshold was established based on evidence that measurements of proteins with this magnitude of change were found reproducible and reliable, and with a high correlation rate between MS and western blot signals (Pearson's $r=0.8-1$) 40.

9. The PDX models were from severely compromised immunodeficient mice, which were significantly different from the human cancer microenvironment. The authors analyzed some mouse proteome features (defined them as stromal) among the proteotypes in their manuscript. However, this mouse proteome is fundamentally different from real stroma in human. I do not think such analysis could reflect the real human tumor microenvironment.

We agree with the reviewer's comment that the tumor microenvironment of the NOD-SCID mice is different than that of patients. This is a limitation of our model that we further acknowledge in our revised discussion as follows:

An obvious drawback of our method was the use of subcutaneous implantation of patient tumors in NOD-SCID mice that have a compromised immune system. NOD-SCID mice have a reduced innate immunity and nearly no adaptive immunity^{74,75}. Although, these immunodeficient features are necessary to prevent tumor rejection, this limits the scope of stromal differences that would usually be present in an immune-healthy individual. This also makes NOD-SCIDs not the ideal model for testing of immuno-modulatory treatments.

While these drawbacks exist, the intact part of the immune system of these models and the murine origin of the stromal component, allows us to clearly see differences among two of our proteotypes, LUAD1 and LUAD3, which highlight the influence of tumor signaling can have on modulating immune components of the tumor cells or of the mouse.

Reviewers' Comments:

Reviewer #1:

Remarks to the Author:

I appreciate the authors' efforts to address my suggestions and the amount of work that has gone into the revision. However, I still believe that the identification of "unique proteotypes", as stated in the title of the manuscript, should be accompanied by a discovery-and-validation approach in terms of NEW actionable vulnerabilities. In this revised version, the information about the value of blocking active targets, as identified in the proteomic survey, is limited to two questionable instances: i) response to the EGFR inhibitor afatinib in a PDX harbouring EGFR amplification (something expected, based on several reports from the clinic; see, among others, Toffalorio et al, J Thorac Oncol 10:392-396, 2015, PMID: 25611230); ii) lack of response to the FGFR inhibitor BGJ398 in PDXs with FGFR1 amplification in the absence of protein overexpression (again, this is not a new piece of evidence: see, for example, Aggarwal et al, J Thorac Oncol 14:1847-1852, 2019, PMID: 31195180; Bogatyrova et al, Eur J Cancer 151:136-149, 2021, PMID: 33984662).

By grouping lung cancer cell lines with available proteomic and pharmacologic annotation into proteotypes, the authors propose a series of proteotype-specific sensitivities, including a top set of candidate drugs based on the effect size of their effect on cell line viability (Supplementary Figure 6F). It is felt that a proof-of-concept study in vivo using representative PDXs with relevant proteotypic features should be conducted to improve the conclusiveness and translational relevance of the dataset.

Reviewer #3:

Remarks to the Author:

The authors have responded appropriately to all of the reviewers' comments including providing additional experimental data and substantial editing of the manuscript as requested by the reviewers.

Reviewer #4:

Remarks to the Author:

The authors improved their manuscript according to my comments. However, some key issues are not satisfactorily or convincingly addressed, especially on the technical parts of mass spectrometry data processing and quantification analysis. The details of mass spectrometry data processing are still ambiguous, which is difficult for data quality evaluation and the community reproducibility. They should provide more details to fit the standards of mass spectrometry-based proteomics data report. In addition, the quality control for the PDX samples and phosphotyrosine data need be carefully evaluated. The samples with low human tumor cell composition or little pY sites quantified should be removed prior to data analysis, or more evidence should be provided to justify the reliability of doing so.

1. The PDX proteome is the mixture of human and mouse proteins. The detailed number of proteins belong to human, mouse or shared by human and mouse was now listed in the revised manuscript. However, the authors did not show the details of which kinds of proteins/peptides were used/selected for further data analysis (normalization across samples, missing value, etc.). Without the details, the community can hardly reproduce the results of this manuscript.
2. According to Fig 4B, the human tumor cell composition across samples was ranged between 20-70%. The samples with low human tumor cell composition should be removed before data analysis due to the large interference or noises. For example, multi-omics analysis for large scale clinical tumor tissues were all based on the tumor purity at least higher than 50% in CPTAC's previously studies (Cell. 2020 Jul 9;182(1):200-225; Cell. 2019 Oct 31;179(4):964-983; Cell. 2016 Jul 28;166(3):755-

765.).

3. In the question 4, the authors did not respond to the critical question on how they integrated the proteome data and tyrosine phosphoproteome data from different quantification strategies. The data processing method and the quantitative standard was very different between the TMT labeling and the label-free quantification method.

4. Even though the authors claimed their method and phosphotyrosine result was comparable to other published studies with limited amount of sample resources, the data quality of phosphotyrosine proteome in this study appears to be relatively low, especially for the samples with less than 100 pY sites quantified. The quantification reliability would be significantly interfered by these low quality phosphotyrosine data, such as the results in Fig 5G-H, Fig 6, Fig 7C, and Fig 7F. The samples with few quantified pY sites should be removed before statistical analysis, or the authors need provide evidence to justify the reliability for including these samples.

5. The authors conducted the pharmacological exploration by using their protein signatures in cell line data of DepMap project in the revised manuscript. However, the results acquired in cell line data could not really reflect the potential drug efficacy in the PDX model.

6. Since the authors also agreed that the tumor microenvironment of the NOD-SCID mice was different than that of patients. The results in Fig. 8 on the differential stromal composition of LUAD in PDX samples are not informative or even misleading. The authors should not present the results in this way.

Reviewer #1, expert in PDX models for lung cancer (Remarks to the Author):

I appreciate the authors' efforts to address my suggestions and the amount of work that has gone into the revision. However, I still believe that the identification of "unique proteotypes", as stated in the title of the manuscript, should be accompanied by a discovery-and-validation approach in terms of NEW actionable vulnerabilities. In this revised version, the information about the value of blocking active targets, as identified in the proteomic survey, is limited to two questionable instances: i) response to the EGFR inhibitor afatinib in a PDX harbouring EGFR amplification (something expected, based on several reports from the clinic; see, among others, Toffalorio et al, J Thorac Oncol 10:392-396, 2015, PMID: 25611230); ii) lack of response to the FGFR inhibitor BGJ398 in PDXs with FGFR1 amplification in the absence of protein overexpression (again, this is not a new piece of evidence: see, for example, Aggarwal et al, J Thorac Oncol 14:1847-1852, 2019, PMID: 31195180; Bogatyrova et al, Eur J Cancer 151:136-149, 2021, PMID: 33984662).

By grouping lung cancer cell lines with available proteomic and pharmacologic annotation into proteotypes, the authors propose a series of proteotype-specific sensitivities, including a top set of candidate drugs based on the effect size of their effect on cell line viability (Supplementary Figure 6F). It is felt that a proof-of-concept study in vivo using representative PDXs with relevant proteotypic features should be conducted to improve the conclusiveness and translational relevance of the dataset.

Response: Our most significant discovery, captured accurately in our title was the discovery of novel proteotypes with prognostic impact. Our findings support our overarching contention that in the future NSCLC patients may be stratified and treated according to proteotype. No other published report on NSCLC primary or PDX models has been able to draw such an impactful conclusion.

However, Reviewer #1 has requested that we complete pre-clinical testing to validate new targets in the PDX models, which was not the purpose of our study. We appreciate the reviewer's request on new studies *in vivo* using our models, but with due respect, we feel such request is truly excessive, unreasonable and unfair, as a truly impactful report of such studies requires the scale that will be worthy of a separate new manuscript and will not fit within the limits of the current manuscript. We strongly believe that conducting large-scale drug screens on PDXs should be a multi-institutional collaborative effort among lung cancer researchers, probably at the scale of TCGA and requiring a large amount of new funding. As far as we are aware, and we are ready to be corrected otherwise, our manuscript constitutes the largest multi-omic profiling data on a NSCLC PDX cohort, including the first such report on detailed proteomics profiling and analysis of the proteome of these tumors, similar in scale (sample number) to those recently published using patient samples. The results are novel including proteotypes with survival differences among the already more aggressive (i.e. engrafting) group among other interesting findings. These proteotypes were further validated in multiple external cohorts and potential targets were predicted, which will be focus of future studies. We also wish to point out that in the multi-omic studies on lung cancer cohorts published recently in *Nature Communications* (PMID: 31395880) and *Cell* (PMID: 32649877, PMID: 32649874, PMID: 32649875), the investigators also did not provide

functional validation or validation of claimed novel targets. The testing of new therapeutic hypotheses in our PDX models would not provide additional validation or invalidation of the prognostic impact of our newly discovered proteotypes. Therefore, we believe the additional works requested are outside of the scope of this study.

We agree with Reviewer #1's recognition of the important need for new treatments in NSCLC. However, the purpose of our study was to deepen our understanding of NSCLC biology through the generation and characterization of a large PDX collection that we expect will help to facilitate the formulation and testing of new therapeutic hypotheses. In this light, we have revised the manuscript to ensure we have not overstated the identification or validation of identified candidate targets. In the revised manuscript we have deleted references to "candidate targets and drug sensitivities" from the Abstract and Introduction and added a concluding sentence to the introduction: "These findings represent new insights into lung cancer biology and suggest that the further characterization of NSCLC proteotypes in NSCLC models including our PDX collection may be an approach to test emerging therapeutic hypotheses."

Reviewer #3, expert in lung cancer genomics/subtypes and therapeutics (Remarks to the Author):

The authors have responded appropriately to all of the reviewers' comments including providing additional experimental data and substantial editing of the manuscript as requested by the reviewers.

Comment: We greatly respect the reviewer's understanding and appreciation of our work and manuscript.

Reviewer #4, expert in proteomics (Remarks to the Author):

The authors improved their manuscript according to my comments. However, some key issues are not satisfactorily or convincingly addressed, especially on the technical parts of mass spectrometry data processing and quantification analysis. The details of mass spectrometry data processing are still ambiguous, which is difficult for data quality evaluation and the community reproducibility. They should provide more details to fit the standards of mass spectrometry-based proteomics data report. In addition, the quality control for the PDX samples and phosphotyrosine data need be carefully evaluated. The samples with low human tumor cell composition or little pY sites quantified should be removed prior to data analysis, or more evidence should be provided to justify the reliability of doing so.

Response to overall remark:

With due respect, we were truly surprised by the reviewer's comment that the level of technical details regarding proteome data is ambiguous and does not fit "standards of mass spectrometry-based proteomic data report." To justify that our reported data do indeed meet or surpass current standards, we have broken down these technical details into several categories (proteome measurement technique, data quality assurance, normalization methodology, report of number of identified proteins, and report of data exclusion from analysis) and compare our presentation of this information to the 3 *Cell* papers referred to by this reviewer (2020 182:200-25; 179:964-83; 166:755-65.). To summarize, the level of technical details shared in our manuscript is either on par or more extensive than these other papers. Please see table below for a detailed comparison:

A. Proteome measurement technique:

Mirhadi et al. (the manuscript under review)	Zhang 2016	Clark 2019	Gillette 2020
In main text: Tandem mass tag (TMT)-based quantitative MS analysis of PDX tumors was undertaken (Supplementary Fig. 2A). Further extensive details regarding tissue homogenization, trypsinization, labeling, HPLC fractionation and MS parameters are disclosed in methods section.	Proteomics measurements used isobaric tags for relative and absolute quantitation (iTRAQ; Ross et al., 2004) in conjunction with offline liquid chromatography fractionation via high-pH reverse-phase liquid chromatography (RPLC) and online RPLC with high-resolution tandem MS to provide broad coverage for peptide and protein identification and quantification.	Technique used not disclosed in main text. In methods: Desalted peptides from each sample were labeled with 10-plex TMT (Tandem Mass Tag) reagents (Thermo Fisher Scientific).	Tandem mass tags (TMT)-based isobaric labeling was used for precise relative quantification of proteins, phosphosites, and acetyl sites.
Conclusion: All 3 CPTAC papers, like ours, are very brief in describing the proteome-based technique used in their study. Zhang et al., 2016 shared their fractionation method in the main text whereas in our manuscript as well as Clark 2019 and Gillette 2020 shared these details in Methods.			

B. Data Quality Assurance

Mirhadi et al. (the manuscript under review)	Zhang 2016	Clark 2019	Gillette 2020
For data quality assurance, a replicate sample pair in the same experimental group and two pairs of replicates samples split into different experimental groups were analyzed. The technical replicates provided a readout for fidelity of the normalization method and technical robustness. A strong linear relationship between the replicates was seen for each pair ($R^2 \geq 0.94$) (Supplementary Fig. 2E). PCA verified that samples did not cluster based on experimental group or isobaric labels (Supplementary Fig. 2F-G).	We used clustering, principal-component analysis (PCA) and statistical tests to identify any significant batch effects associated with the site of analysis (a detailed comparison of within-site, between-site, and between-sample measurement variability and the process used to merge the JHU and PNNL data are given in Figure S1). As shown in Figure S1C, the median coefficient of variation (CV) between measurements at the two sites was 16%.	No quality control results/plots are reported.	Excellent reproducibility and data quality were maintained across the entire dataset (Figures S1C– S1F).
Conclusion: In terms of data quality assurance, our study was more extensive in terms of study design, presentation, and explanation. Our study, in addition to having the control references which the other studies had, also had tumor replicates ran in different TMT groups to ensure data quality. These technical replicates clustered together and had a Pearson r of 0.99. Gillette et al 2020 study only briefly mentions that they had “excellent data quality” and the figures they presented to show this show a technical rep mean with Pearson r of 0.91, lower than ours. They showed this using the reference samples, which are used to normalize and are expected to have very similar values. Clark et al., 2019 did not discuss data quality assurance and			

Zhang et al., 2016 data quality were only concerning the lack of batch affect between the two hospitals the samples were collected from.

C. Normalization

Mirhadi et al. (manuscript under review)	Zhang 2016	Clark 2019	Gillette 2020
Total Proteome Data Normalization Intra-TMT experiment group normalization: samples were normalized to the sample with the maximum sum intensity of each TMT experiment group. Briefly, sum intensity of all protein for each sample was measured. The sample with the maximum intensity in each TMT group was identified. A conversion factor is calculated which is then multiplied by all proteins of that sample. Inter-TMT experiment group normalization: to normalize TMT groups to each other, internal reference scaling method was used as previously described ⁹². Briefly, the control channels containing the pool of tumors (channel 126 and 131 of each batch) within each TMT experiment were averaged and used to create reference values per protein per each batch. The reference values for each protein in each TMT group were then averaged (geometric mean), and scaling factors calculated for each protein to adjust its reference value to the geometric mean value was measured. These scaling factors were then used to adjust the summed reporter ion intensities for each protein in the remaining eight experimental samples in each TMT experiment. For tumor/stroma content normalization, briefly the sum intensity of human proteins was calculated, then the average of these sums was measured. A conversion factor was then calculated to equalize total human protein signal across samples. Same approach was used to equalize the stromal content among PDX samples with mouse proteins. Refer to Supplementary Table 8 for fully normalized Table.	We used the relative abundance measurements for each protein in the 32 patient samples analyzed at both JHU and PNNL to normalize across the two analysis sites	Before performing any downstream analysis, we applied batch correction on global and phosphoproteome abundance to remove the technical difference between different TMT 10-plexes. An R tool, ComBat, with tumor/normal status adjustment was applied to remove batch effects (Johnson et al., 2007).	It was assumed that for every sample there would be a set of unregulated proteins or phosphosites that have abundance comparable to the common reference (CR) sample. In the normalized sample, these proteins, phosphosites, or acetylsites should have a log TMT ratio centered at zero. In addition, there were proteins, phosphosites, and acetylsites that were either up- or downregulated compared to the CR. A normalization scheme was employed that attempted to identify the unregulated proteins phosphosites or acetylsites, and centered the distribution of these log-ratios around zero in order to nullify the effect of differential protein loading and/or systematic MS variation. A 2-component Gaussian mixture model-based normalization algorithm was used to achieve this effect. The two Gaussians ($m_i; s_i$) and ($m_j; s_j$) for a sample i were fitted and used in the normalization process as follows: the mode m_i of the log-ratio distribution was determined for each sample using kernel density estimation with a Gaussian kernel and Shafer-Jones bandwidth. A two-component Gaussian mixture model was then fit with the mean of both Gaussians constrained to be m_i, i.e., $m_1 = m_2 = m_i$. The Gaussian with the smaller estimated standard deviation $s_i = \min(s_1, s_2)$ was assumed to represent the unregulated component of proteins/phosphosites/acetylsites, and was used to normalize the sample. The sample was standardized using (m_i), by

			subtracting the mean mi from each protein/phosphosite/acetylsite and dividing by the standard deviation.
Conclusion: In terms of description of the normalization methodology used, Zhang 2016 and Clark 2019 were very brief. Gillette et al, explains their methodology although some explanations remain vague. We employed a 3-step normalization strategy which is fully detailed in the methods section.			

D. Reporting of quantified proteins

Mirhadi et al. (the manuscript under review)	Zhang 2016	Clark 2019	Gillette 2020
MS analysis of 133 PDX samples uncovered a total of 13284 proteins using a strict false discovery rate (FDR) of 0.01 of which 6830 were identified as human, 4423 as mouse, and 2031 that did not contain unique human or mouse peptides and therefore were assigned as human/mouse (Fig. 4A) (Supplementary Table 8).	A total of 9,600 proteins were identified with high confidence in all tumors, and the relative abundances in each tumor are given in Table S2.	Proteomics and phosphoproteomics analyses identified a total of 11,355 proteins and 42,889 phosphopeptides, respectively, of which 7,150 proteins and 20,976 phosphopeptides were quantified across all samples (STAR Methods).	Number of identified proteins not reported.
Conclusion: In reporting identified proteins we are more detailed than the 3 CPTAC papers.			

E. Exclusion of data from downstream analysis

Mirhadi et al. (the manuscript under review)	Zhang 2016	Clark 2019	Gillette 2020
Proteome hierarchical clustering Proteome hierarchical clustering was performed by using the subset of tumor (human) proteome quantified in at least 70% of samples in the clustering (Fig. 4C, Fig. 5A-B) and using the subset of stromal (mouse) proteome quantified in at least 70% of samples in the clustering (Fig. 7A-C). Protein expressions are log2 transformed and z-score across respective samples. For sample (column-wise) and protein (row-wise) clustering, Pearson correlation distance with average linkage was used using Perseus software default parameters. Principal component and volcano analysis Only a subset of tumor proteome quantified in all samples were used for principal component analysis. Only the two components with the highest proportion of variance were picked for plotting the PCA plot using Perseus software. Volcano analysis was performed for proteins with detection in at least 70% of samples with indicated	Functional analyses and proteome-transcriptome associations were restricted to 3,586 proteins observed and quantified in all 169 HGSC samples used for protein functional analyses and where sample variability (signal) exceeded technical variability (noise) in the merged data	The 3,567 (50%) most variable global proteins without missing values were analyzed by CancerSubtypes (Xu et al., 2017) for consensus clustering (Monti et al., 2003) of tumor subtypes. Specifically, 80% of the original sample pool was	To ensure that poor quality or questionable samples were not included in the final dataset, we performed principal component analysis (PCA) on the RNA-seq, global proteome and phosphosite expression data. In the input to PCA (Figure 7A), we excluded any genes, proteins and phosphosites (in the respective datasets) missing in 50% or more of the samples. For each dataset, we plotted the 95% confidence ellipse in the PC1 versus PC2 plot for the tumor and normal groups. Any samples falling outside

FDR cut offs and $s_0=0.1$ using Perseus software. Analysis of Differentially Expressed Proteome and Pathway Permutation based FDR corrected two tailed student's t-test ($q\text{-value}<0.05$) between one subtype compared to others in that histology type was performed on the entire proteome (human, mouse and ambiguous) (Supplementary Table 9). These proteins along with associated experimental expression and q-values were inputted in ingenuity pathway analysis⁹⁹. Resulting enriched pathways (Supplementary Table 9) were further filtered based on significance $p\text{-value}<0.05$ and activity score as determined by IPA $z\text{-score}>0$ or <0 (Fig. 6A-B, Fig. 8E).	(Table S2), calculated as described in the Supplemental Experimental Procedures.	randomly subsampled without replacement and partitioned into three major clusters using hierarchical clustering, which was repeated 500 times (Wilkerson and Hayes, 2010).	these ellipses were deemed to be outliers. Samples that were outliers in all three datasets (RNA-seq, proteome and phosphosite) and had inconsistent pathology reviews were excluded. Only sample C3N.00545 satisfied all exclusion criteria and was removed from the final dataset.
Conclusion: Our study reports which proteins are used for every downstream analysis in both the Methods section and the main text. Zhang 2016, reports that all subsequent analysis was limited to proteins identified across all samples. Clark 2019 only shares their exclusion criteria regarding consensus clustering and not other subsequent analysis. Gillette et al. only mention their criteria for excluding one of their samples but does not disclose which proteins were used for each subsequent analysis. The information we have provided is more transparent than these other studies.			

1. The PDX proteome is the mixture of human and mouse proteins. The detailed number of proteins belong to human, mouse or shared by human and mouse was now listed in the revised manuscript. However, the authors did not show the details of which kinds of proteins/peptides were used/selected for further data analysis (normalization across samples, missing value, etc.). Without the details, the community can hardly reproduce the results of this manuscript.

Response: We appreciate Reviewer #4's efforts to consider our manuscript. However, we regret that Reviewer #4 has missed these points that were in fact included in the method section of original manuscript, which were then moved to the main text based on this reviewer's comments.

Firstly, the number of proteins (Human, Mouse and Human/Mouse) **was** included in the original submission and was in fact one of our main figure panels (Fig 4A). Regarding the comment on sharing relevant technical details disclosing which proteins were used for subsequent analyses, all such details were included in the methods section of our first submission. Given the comments we received from Reviewer #4 in the first round of revision, we moved all relevant technical details necessary to reproduce the results into the main text of the manuscript. For any subsequent downstream analysis, we have disclosed details of which proteins (human, mouse, or human/mouse) and our missing value cut-offs. These details were and are included in the manuscript. See for example:

"Unsupervised hierarchical clustering based on human/tumor proteins identified in at least 70% of 133 PDX tumors revealed ... (Fig. 4C)."

"Non-murine proteins detected in at least 70% of samples were compared between all LUAD and LUSC samples (FDR< 0.001) (Error! Reference source not found.4D)."

"The subset of PDX human tumor proteome proteins identified in at least 70% of cases, was subjected to unsupervised consensus clustering."

“PCA of tumor/human proteome identified in all PDXs further supports the identification of three distinct proteotypes (Supplementary Fig. 3E), designated LUAD1, LUAD2 and LUAD3.” And so on....

2. According to Fig 4B, the human tumor cell composition across samples was ranged between 20-70%. The samples with low human tumor cell composition should be removed before data analysis due to the large interference or noises. For example, multi-omics analysis for large scale clinical tumor tissues were all based on the tumor purity at least higher than 50% in CPTAC’s previously studies (Cell. 2020 Jul 9;182(1):200-225; Cell. 2019 Oct 31;179(4):964-983; Cell. 2016 Jul 28;166(3):755-765.).

Response: We regret that the reviewer has misinterpreted our results. All tumors had cellularity greater than 50%. Indeed, using histological approaches used in the cited CPTAC studies, a typical PDX has >70% tumor cellularity. The 20-70% signal is referring to the total sum intensity of human proteins, which is not identifiable by all human bulk tumor samples in studies referred to by Reviewer 4.

As written in the manuscript: “To assess tumor-stroma composition, the fraction of total ion intensity corresponding to human, mouse, and human/mouse proteins was determined for each PDX sample (Fig. 4B) (Supplementary Table 8). This provided a unique opportunity to correct for discrepancies in tumor (i.e. human) cell composition across samples, which ranged between 20-70% (Fig. 4B) (Supplementary Table 8).”

Further, we would like to remind that a strong feature of our study is the fact that we were able to discern stromal components from tumor cells. This is because during serial passage in the mouse the human stroma becomes replaced by murine components. Consequently, our cross-species, MS analysis of human-tumor vs. murine-stroma allows us to account for the tumor/ stromal signals before further analysis and hence there was no need to exclude samples.

To demonstrate that our normalization was effective, consider PHLC113-X2, which had the lowest tumor-human proteome signal across all PDXs in our cohort at 22.5%. Interestingly, this model was a passage 5 PDX and we also had PHLC113-X1 in our cohort, which is a passage 3 sample of the same primary tumor with 63% tumor/human signal. The stromal-mouse content of this PDX has increased with increased serial passaged, which is an expected phenomenon. Once the normalization to equalize tumor signal is performed you can see that these 2 samples cluster together among LUSCs, supporting that (1) our normalization strategy is effective, and (2) because of our ability to perform this normalization, tumors with low human signal can remain in cohort for further analysis.

3. In the question 4, the authors did not respond to the critical question on how they integrated the proteome data and tyrosine phosphoproteome data from different quantification strategies. The data processing method and the quantitative standard was very different between the TMT labeling and the label-free quantification method.

Respectfully, the fact is that we did not integrate the proteome and phosphotyrosine data and never claimed we did. A fully integrative analysis would have required integration of quantified kinase/phosphatase and cognate peptide-phosphopeptide levels, such as we have reported previously,¹ but which were not attempted

¹ Karisch R et al. (2011) Global proteomic assessment of the classical protein-tyrosine phosphatome and "Redoxome". *Cell* 146:826-40 PMID: 21884940; Tong J et al. (2017) Integrated analysis of proteome, phosphotyrosine-proteome, tyrosine-kinome, and tyrosine-phosphatome in acute leukemia. *Proteomics* 17:10.1002, PMID: 28176486; Jin LL et al. (2010) Measurement of protein phosphorylation stoichiometry by SRM mass spectrometry. *J Proteome Res* 9:2752-61, PMID: 20205385

in this study. In the manuscript, the phosphotyrosine analysis was independently performed in a supervised manner, by comparing pY-peptides that had been sorted according to proteotype assignment.

4. Even though the authors claimed their method and phosphotyrosine result was comparable to other published studies with limited amount of sample resources, the data quality of phosphotyrosine proteome in this study appears to be relatively low, especially for the samples with less than 100 pY sites quantified. The quantification reliability would be significantly interfered by these low quality phosphotyrosine data, such as the results in Fig 5G-H, Fig 6, Fig 7C, and Fig 7F. The samples with few quantified pY sites should be removed before statistical analysis, or the authors need provide evidence to justify the reliability for including these samples.

Response: Respectfully, we remind that in our original response to the first round of reviews we provided a comparison of our results with comparable published studies that refute this statement. Regarding the suggestion that 'low quality' data for samples with lower identifications should be removed from analyses, we disagree because this would introduce bias in our analysis. Such filtering was not done in similar published studies (PMID: 18083107, PMID: 25670172, PMID: 26356563). In addition, we did not attempt analyses that would be affected by samples with fewer identifications. Instead, pY data were used in supervised analyses, i.e. after stratification according to proteotype. These included: supervised clustering (Figure 5G-H); differential analysis among the groups, which did not include samples with no value (Figure 6C-D); and in Figure 7C we are simply looking at EGFR pY site signals. Lastly, we note that there is no Figure 7F.

5. The authors conducted the pharmacological exploration by using their protein signatures in cell line data of DepMap project in the revised manuscript. However, the results acquired in cell line data could not really reflect the potential drug efficacy in the PDX model.

Firstly, we note that the fact that the DepMap LUAD and LUSC cell lines were effectively stratified by using our proteotype signatures was an overlooked interesting finding. Secondly, we did not claim that the DepMap analysis would "reflect the potential drug efficacy in the PDX model," which would have been an over interpretation that we do not agree with. Rather, we simply noted that the gene suppression vulnerabilities associated with the proteotype-organized DepMap lines was aligned with our proteomics analyses of activated pathways and processes performed on the PDX models. We believe this is a legitimate application of the DepMap utility to interrogate our PDX data for candidate vulnerabilities.

6. Since the authors also agreed that the tumor microenvironment of the NOD-SCID mice was different than that of patients. The results in Fig. 8 on the differential stromal composition of LUAD in PDX samples are not informative or even misleading. The authors should not present the results in this way.

We agree that no cancer model is 100% accurate in its recapitulation of primary tissue and agree with the fact that NOD-SCID mice are immune compromised. However, our results clearly show that the intact aspects of the immune system in these mice are showing distinct proteome differences between two of the LUAD proteotypes. We have reported but not overstated this observation as follows: "Interestingly, LUAD1 and LUAD3 had distinctive stromal proteomes (Fig. 8C), with LUAD3 significantly associating with cluster iii (Fisher exact t-test $p < 0.00001$) and LUAD1 significantly associating with cluster i and ii (Fisher exact t-test $p < 0.00001$). LUAD2 did not significantly correlate with either of these clusters (Fisher exact t-test $p > 0.05$), suggesting it does not establish or maintain a distinct stromal composition." Our observation is meaningful as it indicates that the two different proteotypes recruit distinct stromal components. In fact, the relevance of and validation for our observations have already been demonstrated in patients in Gillette et al 2020, where they reported two immune types among their LAUD samples 'hot' and 'cold'. In our comparative analysis, in the Discussion section, we noted that these two LAUD proteotypes overlap with what was considered as hot vs cold in the Gillette et. al. study as follows: "In LUAD patients, immune "hot" and "cold" subtypes were recently described¹³, where the hot subtype was identified by their stronger signature for B and T cells and macrophages, while also presenting stronger signatures for immune inhibitory cells and processes¹³. We

observed that the hot subtype corresponds to the LUAD3 proteotype. This illustrates the potential for the PDX system to reveal immune system dynamics and potential therapeutic opportunities.”

Reviewers' Comments:

Reviewer #4:

Remarks to the Author:

In the response, the authors made a direct comparison between their methods and CPTAC methods. Indeed, there is fundamental difference between the proteomics dataset in their study and those of CPTAC. Different from the pure human samples that CPTAC used, PDX samples used in this study are mixtures of human and mouse proteins. The method for tyrosine phosphoproteome is different from CPTAC, either. Key information in this study, particularly in data analysis, is missing.

-This manuscript lacks the detail how to define human unique, mouse unique or human/mouse proteins. Mass spec data only provide the information of peptide sequences rather than intact protein sequences. In many cases, there could be human unique, mouse unique or human/mouse shared peptides that are assigned to the same "protein". What if both human and mouse unique peptides belonging to the same protein were identified? This is key information for any further data analysis.

-There is no detail for "tumor/stroma content normalization (line 735)". Why the authors "average" the intensity sums? How the "conversion factor" was calculated?

-In supplementary fig 2B-D, how did the authors calculate the correlation values between mRNA and protein? It is unclear how the authors selected the genes/proteins for CNV-RNA, CNV-protein and RNA-protein correlation analysis. Were human unique, mouse unique or human/mouse proteins were used in these analyses? Or other factors were considered for protein/gene selection?

-It is unclear whether human unique, mouse unique or human/mouse pY peptides were used for tyrosine phosphorylation analysis. There is a lack of detail for the data analysis in Supplementary Fig. 2H-I. What is the exact "maximum signal" used (Line 253)? What conclusion could be made by such analysis? It is not clear how "differential" analysis was conducted for tyrosine phosphorylation data, either. Was any statistical method used?

-The authors only used fold-change as cutoff for lung cancer biomarker analysis, which is a lack of statistical analysis (line 286).

- The authors mentioned "a typical PDX has >70% tumor cellularity" in their response. The tumor cellularity of each PDX need be provided in the manuscript to avoid confusion.

- The tumor microenvironment of the NOD-SCID mice was fundamentally different from cancer patients. Since NOD-SCID mice is immunodeficient, I'm not convinced by their conclusion that PDX system has the potential to reveal immune system dynamics (line 470).

In addition, there are some other issues.

-The authors described they used orbitrap analyzer to detect the CID fragmented ions, which is unable to be realized on an Orbitrap Fusion Lumos (line 669).

-There are some inconsistencies to present the data. For example, a total of 58 LUAD PDX models were acquired according to fig 2A, but the authors claimed to use 59 LUAD PDX samples for consensus clustering (line 229).

REVIEWER COMMENTS

Reviewer #4 (Remarks to the Author):

In the response, the authors made a direct comparison between their methods and CPTAC methods. Indeed, there is fundamental difference between the proteomics dataset in their study and those of CPTAC. Different from the pure human samples that CPTAC used, PDX samples used in this study are mixtures of human and mouse proteins. The method for tyrosine phosphoproteome is different from CPTAC, either. Key information in this study, particularly in data analysis, is missing.

We would like to share that the purpose of the comparison we prepared was not to compare methodology but rather to compare the level of details shared on different aspects of methodology based on the reviewer's comment that the technical details in our study were not on par with the community.

1) This manuscript lacks the detail how to define human unique, mouse unique or human/mouse proteins. Mass spec data only provide the information of peptide sequences rather than intact protein sequences. In many cases, there could be human unique, mouse unique or human/mouse shared peptides that are assigned to the same "protein". What if both human and mouse unique peptides belonging to the same protein were identified? This is key information for any further data analysis.

We thank the reviewer for this important question that addresses an important aspect of our data analysis, and we agree this needs to be more clear for the reader. As stated on page 31 in our manuscript, we utilized MaxQuant's 'Unique+Razor' strategy to deal with quantifying the mixed species data stemming from our PDX analysis. We were remiss in not fully describing in sufficient detail how MaxQuant handles these types of mixed species samples. To address this concern, we have added a more thorough description of this analysis protocol to the methods section page 31 as follows:

"There is a high degree of sequence redundancy in the proteome. In bottom-up proteomics, this leads to situations where often peptides cannot be uniquely associated with one protein of origin. This issue is further complicated in mixed species PDX samples where some shared peptide sequences are identical in human and mouse. In MaxQuant's 'Unique+Razor' strategy, this complexity is addressed by using unique peptides to form distinct protein groups (i.e. human only or mouse only) and with razor (i.e. shared) peptides contributing only to the protein group with the greater number of peptide identifications (PMID: 27809316). In situations where there are no unique peptides for a protein, the shared-peptides are still used to form a protein group but since the specie-of-origin is ambiguous, they are designated as Human/Mouse."

This strategy leads to 4 situations:

Human Unique Peptides	Mouse Unique Peptides	How Razor peptides are assigned to a protein group:
Yes	Yes	Unique peptides used to report a quantification value for each human and mouse protein. Razor peptides contribute only to the protein group with largest number of peptide identifications.
Yes	No	Unique and Razor peptides all used to report a value for Human protein group; no mouse reported.
No	Yes	Unique and Razor peptides all used to report a value for Mouse protein group; no human reported.
No	No	All razor peptides are used to report a Human/Mouse value for protein

2) There is no detail for “tumor/stroma content normalization (line 735)”. Why the authors “average” the intensity sums? How the “conversion factor” was calculated?

We thank the reviewer for this comment and pointing out a lack of clarity. To address this, we have modified our method section to describe the tumor/stroma normalization in more detail as follows:

“For tumor/stroma content normalization, To normalize tumor/stroma content, two conversion factors, one for human-specific proteins and one for murine-specific proteins, were calculated for each sample. To calculate each conversion factor, the sum of total intensities of human-specific proteins for all samples was divided by the number of samples. Then, for each sample, this average value was divided by the sum of total intensities of human-specific proteins for that sample, yielding its conversion factor. The same strategy was employed to calculate a conversion factor for mouse-specific proteins. Normalized protein group values were calculated as the product of measured intensities times the sample-specific conversion factor. Refer to Supplementary Table 8 for normalized values.”

3) In supplementary fig 2B-D, how did the authors calculate the correlation values between mRNA and protein? It is unclear how the authors selected the genes/proteins for CNV-RNA, CNV-protein and RNA-protein correlation analysis. Were human unique, mouse unique or human/mouse proteins were used in these analyses? Or other factors were considered for protein/gene selection?

We thank the reviewer for pointing out this lack of clarity. We have corrected this shortfall by modifying the manuscript as follows:

“To address the postulate that the proteome is largely unpredictable based on abundance of genes and transcripts, pairwise correlations were made between CNV, mRNA and protein²⁶. These analyses were applied for genes/gene products represented at the protein level, and by using **only human proteins that were quantified in all samples**. The resulting Spearman’s Rho values were positive but low in magnitude. The median Spearman’s Rho was 0.33 for CNV-mRNA, 0.22 for CNV-Protein, and 0.3 for mRNA-Protein (Supplementary Table 8) (Supplementary Fig. 2B-D).”

4) It is unclear whether human unique, mouse unique or human/mouse pY peptides were used for tyrosine phosphorylation analysis. There is a lack of detail for the data analysis in Supplementary Fig. 2H-I. What is the exact “maximum signal” used (Line 253)? What conclusion could be made by such analysis? It is not clear how “differential” analysis was conducted for tyrosine phosphorylation data, either. Was any statistical method used?

We appreciate this comment and regret the confusion surrounding our pY analysis. All pY sites were used for all downstream analyses. This was reasonable since all analyses were done in a supervised manner based on proteotype assignments. The purpose of Supplementary Figure 2H-I was to show a normal distribution for identified pY modifications based on both maximal signal and number of pY peptides, which then allowed us to employ statistical methods that rely on the assumption of a normal distribution. The details about maximal signal calculation and statistical methods used are shared in the revised methods section as follows:

“A total of 564 and 484 pY sites were quantified in LUAD and LUSC samples, respectively. Tyrosine phosphorylation was analyzed in a supervised manner for each proteotype. pY signals for each pY site were divided by the maximum signal measured for that site to present the values in a relative manner compared to the maximum value of 1. Then, the average of relative values for each pY site was used for supervised clustering according to proteotypes. Phosphopeptides significantly different by two-tailed

student t-test (p-value <0.05) in one proteotype compared to the others were determined, and used for Ingenuity Pathway Analysis (Supplementary Table 10)⁹⁹.”

5) The authors only used fold-change as cut-off for lung cancer biomarker analysis, which is a lack of statistical analysis (line 286).

We regret that our analysis protocol was confusing and have revised this section accordingly. To identify subsets of proteotype signature proteins we used a 4-fold cut-off to filter the identified significantly differential expressed proteins (FDR<0.05). This approach considers variance, frequency, and fold change. To make this point clearer, we have reworded the text as follows:

“In order to identify protein signatures that could be used to define the proteotype of primary tumors, we considered only significantly differentially expressed proteins as defined by having a more than 4-fold difference (≥ 4 -fold, FDR<0.05) between proteotypes, and detected in at least 50% of cases (Supplementary Fig. 4) (Supplementary Table 9). This threshold was established based on published evidence that measurements of proteins with this magnitude of change were found reproducible and reliable, and with a high correlation rate between MS and western blot signals (Pearson’s $r=0.8-1$)⁴².”

6) The authors mentioned “a typical PDX has >70% tumor cellularity” in their response. The tumor cellularity of each PDX need be provided in the manuscript to avoid confusion.

To satisfy the request of reviewer we have added a column to Table S8-proteome experimental details, where cellularity of each PDX is indicated. The tumor cellularity has been assessed by Dr. Tsao (co-senior author), who is a practicing pathologist.

7) The tumor microenvironment of the NOD-SCID mice was fundamentally different from cancer patients. Since NOD-SCID mice is immunodeficient, I’m not convinced by their conclusion that PDX system has the potential to reveal immune system dynamics (line 470). In addition, there are some other issues.

We thank the reviewer for pointing out our need to further clarify this important issue. We note that NOD-SCID mice, such as we used in our study have a deficiency in the PRDKC gene and a polymorphism in the SIRPA gene. Consequently, they lack an adaptive immune response, and macrophages are less able to eliminate human cells (Yang et al. 2018, PMID: 29983387). However, innate immune cells including natural killer cells and macrophages are still intact. Even if less efficient in some aspects of their functions, macrophages, monocytes, and NK cells persist and can be recruited to the tumor’s microenvironment in NOD-SCID mice. Our data clearly show that the stromal proteomes of LUAD1 and LUAD3 are discernably different with evidence for activation of acute phase response signaling in LUAD3. However, we agree with the reviewer that our general comment on immune system dynamics was an overstatement based on these specific data and have therefore deleted line 470 in the revised manuscript.

8) The authors described they used orbitrap analyzer to detect the CID fragmented ions, which is unable to be realized on an Orbitrap Fusion Lumos (line 669).

We regret this error and thank the reviewer for pointing this out the mistake in our technical description. Indeed the ion trap detector was used for the CID method, and we have therefore modified the text as follows:

“pY peptides enriched from 125 PDX samples and 3 normal mixed tissues were analyzed by using an Orbitrap Fusion Lumos instrument. Samples were loaded by using EVOSEP tips and analyzed with 44 min MS runs as we have described previously (Krieger et al. 2019, PMID: 30938160). Two separate LC-MS/MS runs were performed on every sample, the first one collected collision-induced dissociation (CID)-MS/MS spectra and the other one collected higher-energy collision dissociation (HCD)-MS/MS spectra. The parameters used for MS data acquisition of CID-MS/MS and HCD-MS/MS spectra were: (1) MS: top speed mode, cycle time = 3 s; scan range (m/z) = 400–2,000; resolution = 60,000; AGC target = 400,000; maximum injection time = 100 ms; MS1 precursor selection range = 700–2,000; included charge states 2–6; dynamic exclusion after n times, n = 1; dynamic exclusion duration = 10 s; precursor priority = most intense; maximum intensity = 1E+20; minimum intensity = 50,000; (2) CID-MS/MS: isolation mode = quadrupole; isolation window = 0.7; collision energy = 35%; detector type = Ion Trap; Ion Trap Scan Rate = Rapid, AGC target = 10,000; maximum injection time = 35 ms; Multistage Activation = True, Neutral loss mass = 97.9763; microscan = 1; (3) HCD-MS/MS: isolation mode = quadrupole; isolation window = 0.7; collision energy = 30%; stepped collision energy (%) = 5; detector type = orbitrap; resolution = 15,000; AGC target = 50,000; maximum injection time = 35 ms; microscan = 1.

9) There are some inconsistencies to present the data. For example, a total of 58 LUAD PDX models were acquired according to fig 2A, but the authors claimed to use 59 LUAD PDX samples for consensus clustering (line 229).

We thank the reviewer for pointing out this typographical error. In fact, we used 58 LUAD PDX for consensus clustering, and this has been corrected in the manuscript as follows:

“Among the 58 LUAD samples, which included two technical repeats, consensus clustering revealed 3 groups with high stability (Fig. 5A, Supplementary Fig. 3A-D).”

Reviewers' Comments:

Reviewer #4:

Remarks to the Author:

In this revised manuscript, the authors have now seriously and directly responded to the key questions regarding the technical details of proteomics data analysis. The improvement of these details is critical for reproducing and evaluating their results for the community. I suggest the authors make additional discussion on the technical limitations of mass spectrometry-based proteomics approach for the study of mixed species samples, such as PDX models, since there is obvious technical unsoundness. This is especially helpful to the readers without the expertise in proteomics technology.

**Response to reviewers' comments
NCOMMS-21-04521D**

Reviewer #4: "In this revised manuscript, the authors have now seriously and directly responded to the key questions regarding the technical details of proteomics data analysis. The improvement of these details is critical for reproducing and evaluating their results for the community. I suggest the authors make additional discussion on the technical limitations of mass spectrometry-based proteomics approach for the study of mixed species samples, such as PDX models, since there is obvious technical unsoundness. This is especially helpful to the readers without the expertise in proteomics technology."

We thank the reviewer for helping us better communicate the technical details of our study. We agree there are technical challenges and limitations associated with the analysis of mixed species samples. To address the reviewer's suggestion, we have added the following statement to the discussion:

The MS-based proteome platform analyzes samples that have been digested into peptides. Consequently, those peptides that differ between mouse and human can be used to support the conclusion that the cognate protein from that species has been identified. However, given the high degree of sequence identity between the two species, shared peptides are frequently seen that cannot be distinguished as mouse or human. In these instances, based on common strategies used in the field, signals from shared peptides may be attributed to the protein with the greater number of species-specific peptide identifications. This approach supports determination of mouse/stroma and human/tumor content in PDX samples. However, for any given protein, complementary analyses such as immunocytochemistry and targeted, quantitative MS may further inform on localization and relative abundance.